# The dataset of walled cities and urban extent in late imperial China in 15[th]-19[th] centuries

**Qiaofeng Xue[1], Xiaobin Jin[1], Yinong Cheng[2], Xuhong Yang[1], and Yinkang Zhou[1]**

[1]School of Geography and Ocean Science, Nanjing University, Nanjing 210023, China

[2]College of History and Archives, Yunnan University, Kunming 650091, China

Correspondence: Xiaobin Jin (jinxb@nju.edu.cn)

**Abstract.** Long-term urban extent data are highly desirable for understanding urban land use patterns. However, urban observation data based on remote sensing are typically confined to recent decades. In this study, we advance in this arena by reconstructing the walled cities for China that extend back from 15[th] century to 19[th] century based on multiple historical documents. Cities in late imperial China (the Ming and the Qing Dynasties, 1368-1911) generally had city walls, and these walls were usually built around the urban built-up area. By restoring the scope of the city walls, it is helpful to explore the urban extend in this period. Firstly, we collected the years of construction or reconstruction of city walls from the historical data. Specifically, the period in which the scope of the city wall keeps unchanged is recorded as a lifetime of it. Secondly, specialization of the scope of the city wall could be conducted based on the urban morphology method, and variety of documentation, including the historical literature materials, the military topographic maps of the first half of the 20[th] century, and the remote sensing images of the 1970s. Correlation and integration of the lifetime and the spatial data would produce China City Wall Areas Dataset (CCWAD) in late imperial. Based on the proximity to the time of most of the city walls, we selected six representative years (i.e., 1400, 1537, 1648, 1708, 1787, and 1866) from CCWAD to produce China Urban Extent Dataset (CUED) in the 15[th]-19[th] centuries. These datasets are available at https://doi.org/10.6084/m9.figshare.14112968.v3 (Xue et al., 2021).

## 1 Introduction

As cities are one of the most obvious phenomena on the Earth surface arising from human activities, human productivity has increased significantly since the industrial revolution, which has led to the expansion of population and the acceleration of urbanization (Mumford, 1968; Sanchez-Rodriguez et al., 2005). The rapidly expanding urban built-up area has serious impacts on regional and global changes by modifying the characteristics of the underlying surface while exacerbating human activities such as fossil fuel combustion (Seto et al., 2012; Rodriguez et al., 2018). With complex interactions happening in global environmental changes, the evolution of urban scale and spatial distribution is an important part of global change research (Solecki et al., 2013; Seto and Ramankutty, 2016; Goldewijk et al., 2017; Bai et al., 2018; Kuang et al., 2021). Long-term data on historical urbanization trends and patterns will be conductive to contextualize the current urbanization, as well as to predict future trajectories on its process. In particularly, China has a history of urban construction for thousands of years, and it is also one of the countries with a relatively fast urbanization process in the world today (Gong et al., 2019; Liu et al., 2020). However, China's industrial revolution did not start until the end of the 19[th] century, while the pattern of cities in late imperial China in the Ming and Qing Dynasties (1368-1911) laid the foundation for Chinese cities in modern time (Skinner, 1977).

The data using for the study in the historical period must take into account the availability and integrity even though there are many methods and indicators to assess the level of urbanization.

The widely used data is the statistical material about the population and area of cities for the study of urbanization before the industrial revolution (Doxiadis, 1970). Significantly, population is an effective indicator of the level of urbanization for most current studies to estimate the historical urbanization levels (Chandler, 1987; Reba et al., 2016; Letk et al., 2020). However, in the case of late imperial China, population is not fully applicable to the study of China's urbanization during the Ming and Qing Dynasties for obvious limitation and flaw on the data when the data on urban population was usually originated from the regional level where it included cities, thus few separate statistics data on the number of urban residents could be found, although the official demographics of China during this period were detailed and generally credible (Ho, 1959; Perkins, 1969; Cao, 2001a). For example, William Skinner (1977) used population as the key indicator to measure the urbanization of China in the 19[th] century. However, since China did not have reliable urban population data until 1953, Skinner had to work backward in time, extrapolating better, more recent data to somewhat earlier dates, and building up a consistent time series culminating with the fairly hard data for 1953. Skinner selected 1893 as the representative year, and created a comprehensive file of over 2,500 data cards designed to cover every city and town. Based on this database of more than 150 attributes (mainly including administrative level, circumference of city wall, postal status, population estimates, trade statistics and steamship or rail traffic), cities were classified. Then, he defined the urban population class intervals that the upper boundary of each class was twice the lower boundary, the following series was used: 1,000, 2,000, 4,000, 8,000, 16,000, 32,000, and so on. And finally, Skinner estimated the urbanization process of China in the 19[th] century. It is acceptable to use data of the 1950s to study the urbanization in the 19[th] century; but for longer-term research, the credibility and operability of this approach will be greatly reduced. In summary, the flaws in the original materials have led to a great controversy over the different versions of estimation on Chinese urban population during this period (Li and Wu, 1997; Cao, 2000; Cao, 2001b).

Another way to explore the urbanization process in the historical period is restoration of the urban extents or the built-up areas of cities (He et al., 2002; Hedefalk, et al., 2019; Lin et al., 2017; Qin et al., 2019; Uhl et al., 2021). However, before the popularization of scientific Cartography in the 20[th] century, maps in China generally lacked the basis of surveying and mapping (Yee et al., 1994; Cheng, 2019), and could not be used to restore the urban built-up areas in late imperial period precisely. In addition, there was a lack of statistical data on urban area in late imperial China. Therefore, researchers generally use alternatives to represent the built-up areas of Chinese cities in late imperial period, and the one of the most commonly used indicator are the scope of city walls (Skinner, 1977; He et al., 2002; Qin et al., 2019).

How can the scope of a city wall represent the urban extent? Here we must begin by attempting to summarize the city wall building history that existed in imperial China. The city wall is considered to be one of the basic symbols of ancient Chinese cities (Chang, 1986). But to be specific, cities in China were not always walled. In addition, the characteristics of city walls in different eras were not the same. During the 3[rd] to 10[th] centuries, small cities in China generally had no walls. Even regional capital cities only built small-scale city walls called *Zi-cheng* (*Zi* means small and *Cheng* means city wall). The *Zi-cheng* was built around the government and military barracks, just like castles in medieval Europe. Residential areas, markets, schools and religious buildings were all outside the *Zi-cheng* (Lu, 2011). From the 10[th] to 13[th] centuries, there were some large-scale city walls built around residential areas, but they were generally confined to

few important cities. During the Mongolian-ruled Yuan Dynasty (13-14[th] centuries), many city walls were deliberately torn down. Only in the Ming and the Qing Dynasties (14-19[th] centuries), cities generally built large-scale walls to protect governments, temples, granaries, residences, and certain natural resources against invasion, tribal uprising, and peasant rebellion. According to many previous studies (Chang, 1970; Kostof, 1992; Knapp, 2000), city walls in this period were usually slightly larger than the built-up area of the city, and as the suburban areas grew, new and larger city walls were often built. Thus, the city wall in the Ming and Qing periods could be regarded as the urban fixation line, which reflected the extent of the city. On the other hand, the Ming period and the first century of the Qing witnessed the extensive construction of city walls. 80% of cities in China had walls in the 15[th] century, and in the 16[th] century, 95% of cities were walled (see the details in Section 5 below). Through the study of the scope of the city wall, it will help to reconstruction the urban extent in the late imperial China in 15-19[th] centuries.

Historical materials in the Ming and the Qing Dynasties in China recorded the length and construction time of the city wall of each administrative city above the county level in detail, which provided reliable information for restoring the scale of the city walls. Researchers have estimated the built-up area of Chinese cities in late imperial period by converting the perimeter of the city wall into the area of the city wall (Skinner, 1977; He et al., 2002; Cheng, 2007). However, due to the shape of the city walls were often irregular and their construction years were different from each other, the mentioned urban built-up area estimation often produces large errors. In addition, the differences between scope of city wall and urban built-up area have not been much discussed. There is still lack of city wall and urban extent datasets with high resolution and definite age of late imperial China.

The aim of this project was to collect multiple historical data related to the city walls of late imperial China, digitize it, and make China City Wall Areas Dataset (CCWAD) and China Urban Extent Dataset (CUED) in late imperial in the 15[th]-19[th] centuries. We used a similar method to product a dataset of urban extent areas in Northwest China in the Ming and the Qing dynasties (Xue et al., 2018). And in this new database, we improved the research methods and extended the study area across China. Firstly, based on the historical urban morphology theory (Conzen, 1969), we restored the scope and construction time of walls of each administrative city in the Ming (1368-1643) and the Qing (1644-1911) dynasties, and made the CCWAD product. Then, we analyzed the years and sites of the construction of the city walls, and we found out six representative years that could illustrate the general level of urban extent in China of this period. Based on this strategy, we developed the product of the CUED product in 1400, 1537, 1648, 1708, 1787, and 1866 across China. These datasets provide a foundation for understanding cities in the traditional agricultural society, and they will also be helpful in current and future research and practices in urban environmental and cultural sustainability.

**2 Study area**

This research aims at the cities in China in 15[th]-19[th] centuries. Definition of city is the same as the general research practice of ancient Chinese cities, namely administrative cities, including *county*, *Zhou*, *Fu*, and *Ting*. In addition, the military cities of the Ming Dynasty, *Wei* and *Suo*, and the *Eight Banner cities of Manchu* of the Qing Dynasty were added.

The research period consisted of the Ming and the Qing Dynasties, and there were some differences in the territory of the two dynasties. In order to explore the temporal and spatial characteristics of late imperial China's urban extent, the study area is divided into five sub-regions

based on landform types, local socio-economic history and ethnic distribution, as shown in Figure. 1. (I) Northeast China, which mainly covers the area to the east of Daxing'anling mountain and the north of the Great Wall of the Ming Dynasty. This region was sparsely populated until the influx of large numbers of immigrants in the 18th-19th century, and a number of cities were established at the end of the 19th century and the beginning of the 20th century. (II) Inner Mongolia, which was to the north of the Great Wall and was inhabited by Mongolian herdsmen in 15th-19th centuries. (III) Traditional Agricultural Area was densely populated, with many cities and a long history. (IV) Xinjiang was located in the continental interior, and the population was concentrated in oasis. It became the territory of the Qing Dynasty after the mid-18th century. (V) Qinghai-Tibet Plateau is mainly located on the Qinghai-Tibet Plateau, which is the highest-elevation plateau in the world. There were some historic cities on the edge of the plateau, but the administrative cities within it were established very late.

**3 Data sources**

**3.1 City wall records in historical literature**

There were detailed and systematic records of city walls in Chinese historical literatures, such as the *Book Integration of Ancient and Modern Times* (edited in 1701-1728), *Unified Records of the Qing Dynasty* (edited in 1842), and more than three thousand *Local Chronicles* edited before 1949 all over China. There was a tradition of compiling *Local Chronicles* in the Ming and Qing Dynasties. Most of these literatures were compiled by local governments, and the city wall, as an important achievement, had been paid much attention. These records detailed the construction and transformation of local city walls, such as their construction time, scale and form (see Figure 2). And the *Book Integration of Ancient and Modern Times* and *Unified Records of the Qing Dynasty* were collections of *Local Chronicles*. The historian in our research team have systematically collated and studied these literatures, and compiled a series of Data Compilations (Cheng, 2016a, 2016b, 2016c). And the historical literatures of this study were from these Data Compilations.

**3.2 Old maps and remote sensing image**

Spatialization of the text of historical data was the next step to make this database. Most of the city walls of Chinese cities were demolished after 1949, which made it impossible for us to spatialize them directly on today's map. Fortunately, the 1: 25,000, 1: 50,000, and 1: 100,000 military topographic maps produced by the bureau of surveying and mapping of the Republic of China (1912-1949) and the Japanese army in 1910s-1930s drawn the location of the city walls, making it easier to restore these walls on modern maps (Figure 3a). These topographic maps were mainly plotted in the periods of 1916-1925 and 1930-1939, and they are mainly collected in Taiwan and Japan at present (Jiang, 2017). More than sixty thousand digitalized maps covering 25 provinces in China can be viewed online on various websites, and an integrated query system has been launched (http://map.rchss.sinica.edu.tw/).

In addition, we also need some remote sensing images for auxiliary work, and the CORONA photographs are the most important. CORONA is the satellite deployed by the United States in 1958, and it takes remote sensing images covering the world from 1960 to 1972. Now the CORONA photographs have been decrypted and can be downloaded from the USGS website (https://earthexplorer.usgs.gov/). Before the 1980s, the city of Chinese mainland has not started large-scale expansion, and the ancient relics can be clearly indentified from these remote sensing images. And the modern remote sensing images are obtained from Google Earth.

**3.3 City sites and their lifetime**

We need obtain information of cities in China during the study period including where they were
located, what time they appeared, and when they disappeared. As mentioned above, the research
object was administrative city. If a site was chosen as a local administrative center, it would be
regarded as the birth of a new city; if all the administrative agencies mentioned above were
abandoned or moved, then it will be regarded as the abandoned city; and the period between them
was called the city's lifetime. Most of the city's lifetime information can be obtained from the
China Historical Geographic Information System (CHGIS, Version 6, 2016; available at
https://dataverse.harvard.edu/dataverse/chgis_v6/). In addition, we supplemented and corrected
some missing and mistaken data of CHGIS based on the *Historical Atlas of China* (Tan et al.,
1982) and *General History of Administrative Regions in China* (Zhou et al., 2007-2016). Through
the above work, the city site point layer of the Ming and Qing Dynasties could be obtained, as
well as the time records they set up or abandoned, including 2,560 lifetime records for 2,376 city
sites in total (Figure 1), functioning as the basis for the next step to make the CCWAD and the
CUED products.

**4 The strategy of developing the CCWAD product**

**4.1 The historical urban morphology theory**

The historical urban morphology theory was proposed by British architect Michael Conzen,
emphasizing the importance on studying the urban plan pattern from the perspective of
morphology (Conzen, 1969). It was believed that the urban plan pattern was a complex record of
the development of urban form, which retaining the residual characteristics of each stage of its
development process. Therefore, based on the evolutionary perspective, it is a worthwhile analysis
method to study and reveal the potential history from the existing planning pattern. The urban
morphology theory focuses on large-scale city map, combine with field research and literature
analysis, to analyze the urban plane pattern based on the perspective of evolution, and interprets it
as three elements complex: street and its layout in the street system; burgage and its agglomeration
in the block; and block-plan of a building. And the city wall are generally considered as an
important "fixation line" that has the role of defining the static edge of the city (Conzen, 1969).

Conzern also put forward a series of basic concepts to describe the urban form and its evolution
phenomenon, which is of great significance to the study of urban historical form in China (Li et al.,
1992; Zhong, 2015; Lai, 2019). Chinese researchers often combine historical text data and old
maps to fix the lack of systematic ancient cadastral records. The main elements of the urban flat
pattern are appropriately adjusted to aggregation including streets, water systems and bridges, city
walls, moats, government offices, and temples for analysis. Thus, a relatively clear urban plan
pattern was obtained on several time sections in the pre-industrialization period. The production of
our database does not involve the restoration of streets and buildings, but focuses on the
restoration of the location of the city walls, thus reducing the difficulty of practice and the
requirements for the fineness of the original materials. With the historical urban morphology
theory, it is not difficult to restore the location of city walls in late imperial China by combining
historical literature data, old maps and remote sensing images with some necessary field
investigations, thus helping to understand the urban extent of this period in China.

Figure 4 provides a schematic overview of dataset construction and is referred to throughout the
methods section to clarify the dataset development process.

**4.2 Restoration of the scope of the city walls**

Sorting out the city wall records in historical records and tabulating them by Microsoft Excel

involved much work on filtering the city wall information in the historical literature data since it is
lengthy, messy, and mixed with many literary descriptions. Besides, the perimeter of the city walls
recorded is often not accurate and can only be used as a reference. Therefore, it is focus on
extracting information about construction time and reconstruction time. The literary descriptions
of city walls in the historical records were helpful to the interpretation of remote sensing images,
and were retained as for reference.

We georeferenced and digitized the military topographic maps and the 1970s remote sensing
images. In the georeferencing process, we used modern topographic web maps and Google Earth
to identify common points in the historic maps and the CORONA photographs, such as temples,
city gates, city walls, drum-towers, and crossroads. Using all of the above processed materials, it
is allowed to identify the location of city wall ruins, or other associated ruins, on the Google Earth.
Then, according to the literary description in historical records, the correspondence between the
text records and the identified ruins are judged, thereby identifying the time of the ruins.

Although most of the city walls of Chinese cities were demolished after 1949, there were still
many associated relics, such as the moat parallel to the city wall, or a ring road built after the city
wall was demolished, as well as the radial spread of multiple roads often implies the location of
the city gate. These associated relics could be investigated from remote sensing images of the
1970s, and even in modern remote sensing images (e.g., see Fig. 3 b, c, d). For example, Figure 5
and 6 show the scope of the city walls of several famous Chinese cities from 1368 to 1911, and the
red lines on these figures are the location of city walls presented in the dataset. The eight cities
shown in Figure 5 did not change the scope of the city walls during the period, while the six cities
in Figure 6 changed to varying degrees. Among these cities, Nanjing in Figure 5 and Xi'an
(1368-1642) in Figure 6 have retained relatively complete city walls today, so it is not difficult to
restore their scopes on the remote sensing images. Chengdu, Hangzhou and Suzhou in Figure 5
retained their city moats, so their city walls were located inside the moats. Shanghai and Kunming
in Figure 5 and Beijing, Shenyang, Tianjin (1369-1860) and Urumqi in Figure 6 demolished their
city wall and built ring roads on its old site, for example the "Second Ring Road" in Beijing and
the "Renmin Road" in Shanghai, so their city walls position overlaps with these ring roads. The
scope of city walls in other cities were verified through various ground markers and Local
Chronicles. In cities where the scope of the city walls changed, most of the newly built walls were
located outside the old city gates (e.g. Xi'an, Lanzhou) or around the old cities (e.g. Shenyang,
Tianjin). This was to protect the newly urban built-up areas. There were also cities that built a new
city wall far from the old city (e.g. Urumqi).

Target geographic objects, such as city walls, city gates, moats, and ring roads built after the
city walls demolished, were digitized as temporal snapshots from the maps. The georeferencing
and    digitalization    steps    were    performed    by    using    ArcGIS    Desktop    10.3
(http://www.esri.com/software/arcgis/arcgis-for-desktop/). It would be next step to generate layers
in .kml format on Google Earth, marking their corresponding lifetime, and then use ArcGIS
Desktop 10.3 to covert .kml layers into .shp format. The .shp layers are associated with the Excel
table that previously saved the Local Chronicles data, thereby generating the .shp layer of the
scope of the city walls area with spatio-temporal attributes.

This section shows the process of making the CCWAD product during the Ming and Qing
Dynasties. Users could query and obtain the nationwide city wall area data for any year during
1368 to 1911 by GIS software from this dataset.

## 5 The urban extent data with the CUED product

Now we attempt to extract urban extent data from CCWAD. It must be emphasized that although city wall could be a helpful indicator for representing the extent of cities, there are always gaps and latencies in both definitions and spatiotemporal changes between the city walls and urban extents. The city wall was a functional building with high cost. And it would be built only when it was of vital importance to military and economic defense. Therefore, the scope of the city wall must be adapted to the physical boundaries of the urban built-up area at that time. However, the urban extent would not remain unchanged forever, it would change accordingly with the increase or decrease of urban residents. In contrast, after the city walls were built, the scope of the city walls generally did not change with the built-up areas over time. The overflowing population would build contiguous settlements outside the wall, especially during periods of peaceful and prosperous periods. And during these periods, the scope of city wall could not be consistent with the urban land use. In addition, the urban boundaries before the construction of the city wall were practically unknown. Finally, some special cities, such as those established in the northeast of China at the end of the Qing Dynasty, and some urban concessions (such as the Shanghai concession) established by foreigners in the 19th century, often did not build city walls.

After considering the relationship between the scope of the city wall and the urban extant, we think that the city wall could be regarded as the urban boundary at least during the period when the city wall exerts its functional role; and the closer the time to the construction of the city wall, the more consistent the scope of city wall and the urban extent. Therefore, as long as the appropriate periods were selected, the scope of city walls in these periods could be very approximately regarded as the urban extent. In small-scale studies, users can refer to the above principles and select proper data from CCWAD, and regard the scope of city walls as the urban extents.

CCWAD may enough to satisfy the demand of local and case studies. However, long-term and large-scale urban extent data are highly desirable for urban studies. Since city wall can be regarded as a helpful indicator of the extent of cities, we hope to provide an acceptable national-scale urban extent dataset based on the CCWAD. This is the China Urban Extent Dataset (CUED). To make CUED, it is necessary to extract some suitable representative years from CCWAD to make the time of city boundaries in close proximity to the time of most of the city walls built. This requires statistics and analysis of the city walls' area, the number of walled cities, and the total number of all cities.

We plotted the time series of the number of city walls built (Fig. 7b), the total number of cities (Fig. 7d), the total number of cities that built the city wall (Fig. 7e), and its percentage of the total number of cities (Fig. 7c). It can be seen from Figure 7b that there were some connection between the number of wall constructions and the area of the walls scope. The periods of more constructions were often of faster area growth, and the less construction periods were always of area decline or unchanged. In 1368, there were 1,375 cities in China, of which 851 had city walls, accounting for only 62% of the total (Fig. 7c, d, e). However, in the year 1393, 70% of cities had city walls; in 1469 it reached 80%, in 1540 it was 90%, and in 1576 it was 95%. Since then, even though the number of cities fluctuated to a considerable extent, the proportion of cities with walls to the total cities has remained stable between 95%-97% for a long time. But after 1868, this percentage began to decline, and after 1900 it dropped sharply.

According to the above facts, we selected six base years where the area of the city wall scope

were closest to the urban boundary from the six time periods (i.e. 1368-1404, 1405-1564, 1565-1662, 1663-1727, 1728-1860, and 1861-1911), to product the CUED product in 15th-19th centuries. The selection criteria for the representative years are as follows. Firstly, the proportion of cities with walls in the total cities should be higher. The proportion should generally be more than 90%, except in the 14th and early 15th centuries. Secondly, after the city walls were built, the scope of the city walls generally did not change with the built-up areas over time, so the representative years should be within only one or two years after the end of a large-scale construction activities of the city wall period. In addition, the representative year should be selected at a moderate level of changes in the scope of the city wall within the period. Finally, the representative year should avoid major political, military events and severe natural disasters in order to reflect the general level of urban development in that period.

Therefore, we selected 1400, 1537, 1648, 1708, 1787, and 1866 from CCWAD as the representative year to develop the CUED product in 15th-19th centuries. In these representative years, the scope of city walls and the urban extent were relatively close at the national level. CUED provides the urban extent data with long-term and national-scale.

**6 The accuracy of the CCWAD and CUED**

**6.1 Accuracy ranking system of the CCWAD and CUED**

Due to the differences in data richness and existing relics in various cities, the accuracy of the scope of city walls would also be different. Reliability is a necessary factor to allow researchers and data users to be aware of the accuracy of the data and the subsequent analytical results. So we established an accuracy ranking system for the entire dataset to test consistency. The accuracy ranking is based on the reliability of restored results. It consists of three accuracy levels, A, B, and C, and two special case marks, D and BW. The accuracy ranking A indicates that the authors are quite certain about the restored result, the B indicates that part of the restoration is speculative, and the C means that the restoration is entirely based on supposition. The accuracy ranking is mainly depends on the richness of the city's historical documents and the integrity of the ground remains. But the accuracy levels are basically subjective decisions of the authors. In addition, the D indicates that the city has never been walled, so its urban extent is entirely speculative. And those of rank BW indicates that the city did not build a city wall during this lifetime, but it was built later (next lifetime). It expresses the speculation on the urban extent before the city built its original city wall. The hypothetical results of C, D and BW were based on the city's limited historical documents and physical remains, its administrative level as well as the size of the nearby cities. All the rankings were determined after discussion by all authors.

In summary, the accuracy ranking A and B are more credible, accounting for 90% of the data of CUED, and 69% of CCWAD. The C and D together account for 5% of CUED and 17% of CCWAD. Limited by objective conditions, the extent of some cities may be difficult to restore, but it may not be appropriate to exclude these cities directly. Although the accuracy ranking is an uncertainty attribute in our dataset, it is created with the intention of allowing researchers to subset the dataset to the most suitable level of accuracy for each specific analysis. For example, for studies where the most exact information is required, cities with a certainty ranking of C or D could be rejected. Therefore, we developed the accuracy rankings so that users with different needs could decide how to use these speculative data. Furthermore, improvement and enhancement of the dataset can be better targeted to those cities where geo-locations are suspect—cities with an accuracy value of B or C.

**6.2 Comparison with existing historical urban land use results**

To validate CCWAD, we use the estimation-based provincial Urban Land Use Data (ULUD) in the Qing Dynasty in China (He et al., 2002). Based on the length of city walls data collected from historical documents, ULUD reckoned the areas of urban land use for 18 provinces in 1820. We extract data for 1820 from CCWAD, and choose the 1820 administrative division data provided by CHGIS (https://dataverse.harvard.edu/dataverse/chgis_v6_1820) to count the area of the scope of city walls in each province. Then we compare the result with the ULUD to validate our dataset (Figure 8). It is found that areas of the scope of city walls from CCWAD in 1820 showed good consistency with the ULUD ($R^2$=0.89), signifying the reliability of our CCWAD products. But the area of the scope of city walls in each province of CCWAD is only about 60% of the ULUD. This is probably subject to the overestimations of the urban area in ULUD since ULUD focus on the length of city walls. The length of city walls recorded in Chinese historical documents is often exaggerated. And ULUD assumes that the shapes of city walls are all square or round, which is far from the actual situation.

**6.3 The relationship with historical urban population**

The increase in urban population is one of the main driving factors for urban land expansion (Paclone, 2001). Thereby we further compared the urban extent data in CUED with the Urban Population Data (UPD) in the Qing Dynasty from Cao (2001b) to validate the accuracy of CUED. UPD provides the urban population for 18 provinces in 1776 and 1893 in the Qing Dynasty, and we count the urban extent areas of these provinces of CUED in 1787 and 1866 for comparison (Figure 9). UPD includes towns, so its subject is slightly more than our CUED. The scatter plot between urban population and urban area shows that, on the whole, urban area increased with the urban population, but they are not linearly dependent. In the late 18[th] century, the urban area and urban population of most provinces are significantly correlated. However, Zhili (today's Hebei, Beijing, Tianjin and northeastern Henan), Shanxi, Shandong and Henan have a higher level of urban area than their urban population. It perhaps because these provinces are close to the capital and the Great Wall, the average size of their city walls is larger. Jiangsu and Zhejiang have a lower level of urban area than their urban population, indicating that the urban population density in these provinces is higher and there are more towns (Figure 9a). In the mid to late 19[th] century, with the increase in foreign economic activities, the urban population density of the southeast coast (Guangdong, Zhejiang, Jiangsu) and the midwest (Sichuan, Hubei) increased significantly (Figure 9b). Long-term changes of the relationship between urban area and urban population are accurately described by CUED, which demonstrated the reliability of CUED.

**7 Results**

Based on the CCWAD product, we plotted the time series of the changes in the area of the city walls scope. Taking the area of the city walls in 1368 (=1,087.06 km$^2$) as the initial value, Figure 7a reflects the changes in the area of the city wall area during the Ming and Qing Dynasties in China. It can be seen that in the 14[th]-20[th] centuries, the scope of the city walls area grown at a slow rate. The smallest area of the city wall was in 1373 (=1,040.98 km$^2$), and the largest area was in 1911 (=1,367.22 km$^2$). According to the change of the slope of the Figure 7a, the area change of the city wall scope can be divided into six periods. Period 1368-1404 was in the early years of the Ming Dynasty, many cities were abandoned due to years of war, which led to a decline of city wall areas. However, these cities were quickly rebuilt as well as many military cities were built, making the built-up area soon exceed the level of 1368. At the beginning of the 15[th] century, the

Ming Dynasty abandoned the area north of the Great Wall, and most of the cities in this area were abandoned. After that, in the period 1405-1564, the city wall scope area grew slowly. Since the middle of the 16th century, the situation in the north and southeast was tense, and many cities there built outer city walls, which accelerated the growth of the city wall scope area (period 1565-1662). In the middle of the 17th century, the city wall scope area fell again, partly because of the war in the late Ming and early Qing dynasties, and also because the Qing government abolished many military cities built by Ming Dynasty (period 1663-1727). The growth of the city wall scope area in the period 1728-1860 was very slow. Until the middle of the 19th century, the government opened up immigrants to the northeast of China, and the city wall scope area began to grow rapidly.

Figure 10 based on the CUED product shows the urban extent areas in some provinces in each representative year. Combine with Table 1 and Figure 1, it could be seen that provinces in the northeast of the Region III had the largest urban extent area in late imperial period in 15[th]-19[th] centuries. Hebei, where the capital Beijing was located, had the largest urban area. Jiangsu and Shanghai, an economically developed area, ranked second, and Henan, a populous province, ranked third. Shandong, Shanxi and Zhejiang also have large urban areas. During the study period, the urban extent of the above provinces increased steadily or slowly, but Zhejiang province decreased slightly in 1708. That was because the Qing Dynasty issued an order to demolish some coastal cities at that time. The urban extents of other provinces in the Region III were roughly the same. Among them, Anhui, Guangxi, Hubei, Hunan, Jiangxi, Sichuan and Chongqing had long history of land development, and the urban extent had remained stable during in 15[th]-19[th] centuries. Fujian, Guangdong and Hainan decreased slightly in 1708 by the same reason with Zhejiang. Yunnan and Guizhou province developed intensively and built a number of cities in the early Ming Dynasty. In the middle and late Ming Dynasty, the urban extent of Shaanxi, Liaoning, Gansu and Ningxia increased rapidly because of the severe military pressure faced by nomads at that time. Taiwan began large-scale development only after the 18[th] century, and some small cities were built mainly on the west coast.

Jilin and Heilongjiang, located in the Region I, had no administrative cities in the Ming Dynasty. After the mid-18[th] century, with the influx of immigrants, a number of cities were established. Inner Mongolia, located in the Region II, had a certain number of cities in the Yuan Dynasty (1271-1368) and the early Ming Dynasty, but by the middle of Ming Dynasty, these cities were gradually abandoned. It was not until the late 18[th] century that Inner Mongolia rebuilt some cities with the influx of immigrants. Xinjiang, located in the Region IV, was not under the rule of the Ming Dynasty. In the late 18th century, the Qing Dynasty completely conquered Xinjiang and established a number of administrative cities. And the cities of Qinghai of the Region V were located in the valleys of the Yellow River and Huangshui River.

**8 Data availability**

The datasets include the CCWAD in 1368-1911 and the CUED in 1400, 1537, 1648, 1708, 1787, and 1866 are publicly available and can be downloaded from https://doi.org/10.6084/m9.figshare.14112968.v3 (Xue et al., 2021).

The CCWAD we provide a shapefiles file (referring to files with .cpg, .shp, .dbf, .shx, .sbn, .sbx, and .prj extensions). Appendix A provides an introduction to the attributes of CCWAD. The CUED we provide six shapefile files (referring to files with .cpg, .shp, .dbf, .shx, .xml, .sbn, .sbx and .prj extensions). Appendix B provides an introduction to the attributes of CUED.

**9 Conclusion and outlook**

Ultimately, we view CCWAD and CUED as a beginning compilation of a richer historical, city-level urban database in late imperial China. Despite of the current reliability gaps, these datasets provide a spatially explicit, long-term historical record of walled cities and urban extent of China especially no alternative geo-coded dataset at such resolution exists. As a result, this dataset could be used as a foundation to build a full and accurate record of urban built-up areas through history, creating systematic, global built-up area data to measure urban growth at a long timescale.

However, we caution potential CCWAD and CUED users of the following limitations and dataset details:

1. The urban extent dataset (CUED) is a derivative of the city wall (CCWAD). Strictly speaking, the scope of the city wall cannot be completely equal to the scope of the urban extent. The data may better reflect the urban extent in which year the city wall was built. The lifetime of each urban extent provided by the CCWAD is a period of time, and the urban extent of any year within the time period can be intercepted. However, if the year of interception is too far from the year of construction of the city wall, the actual urban extent may have a large difference with the wall's scope. Before the construction of the city wall, in fact, we were hardly to know the actual scope of the urban extent, and only the later wall's area was referred to. More often, after the city wall was built, as time goes by, the area farther away from the city gates and the center were gradually becoming uninhabited and even becomes cultivated land; the area with convenient transportation outside the city gates forms new built-up areas. Therefore, we recommend that potential CCWAD users should be careful not to be too far away from the year of construction of the city wall when choosing the research years. And this was why we generated six representative years in the CUED product in $15^{th}$-$19^{th}$ centuries China.

2. In general, the increase or decrease of the city wall range often means the increase or decrease of the urban extent, but they are not completely synchronized in time. Like most ancient civilizations, city walls in China were primarily defensive military structures. In peacetime, the city walls were useless and often hindered the expansion of cities. During these periods, suburbs grew outside the city gates, and the walls were often neglected or even vandalized. But during the war, the walls became necessary facilities to defend the cities. At this time, if the suburbs outside the city gates had grown large, new suburban walls were built to protect them. Therefore, a paradox is that the development of cities generally require peaceful social environment, but the expansion of the city wall area often happened in the period of wars. In this sense, the city wall can be seen as the sign and confirmation of the urban development before wars. Users should understand that it is not the war that has led to the expansion of urban extents, but the expansion of the city wall reflects the development of the city's economy and the increase of population before the outbreak of wars.

3. To sum up, the reliability of this dataset is acceptable, but users need to be aware of whether the reliability rating of the area has fallen when it comes to smaller areas. In the 15th-19th centuries, cities in some regions generally did not built city walls. We use accuracy ranking D to represent the cities without walls in CUED and CCWAD. In CCWAD, there have 436 such kind of cities, accounting for 13%. In CUED, there are 83 such cities in the representative year 1400, 48 in the year 1537, 43 in the year 1648, 31 in the year 1708, 37 in the year 1787, and 42 in the year 1866; and the proportions are between 2% and 5%. Cities without the walls could be roughly

divided into two categories. One was the less important cities located in the inland areas. The other was the cities established at the end of the 19th century. At that time, with the advancement of weapons, the defensive significance of the city wall was greatly reduced. When researching these areas, be sure to pay attention to the reliability rating.

4. The objects of our study only include administrative cities. Although almost all cities in the late imperial China could be classified as administrative cities, we must point out that the following types of settlements could also be regarded as "cities", but they are not included in our datasets. (a) In the late imperial China, the industrial and commercial settlements without administrative agencies were generally called "markets (*shi*)" or "towns (*zhen*)". The size of the town was generally smaller than the lowest administrative center, the county seat. But there were also some huge towns, such as Hankou, Foshan, and Jingde, etc., whose scale exceeded the county seat and even higher-level cities. These huge towns should undoubtedly be regarded as cities, but they are not in scope of this research. (b) If a city was already there, and got chosen later to become an administrative center, in this case, data before the "city" became the administrative center were not included in our datasets. (c) Cities outside the direct administration of the Ming and Qing empires, such as Lhasa. (d) Cities belonging to colonists, such as Macau, Hong Kong, and Qingdao, etc. The definition of "city" or "urban" in the late imperial China is complex and far from conclusive, but we hope that the content of our datasets to have a clear border. Therefore, in this study, we defined "city" as the settlement which the administrative center was located. And this definition is the same as the general research practice of pre-modern China. As for the cities outside the range of this study, further detailed explorations are needed.

**Appendix A: Data records of CCWAD**

The China City Wall Areas Dataset (CCWAD) in 1368-1911 we provide a shapefile file (referring to files with .cpg, .shp, .dbf, .shx, .sbn, .sbx, and .prj extensions). It includes the following attributes:

| | |
|---|---|
| FID | The (unique) identifier for each object (integer). |
| NAME | The longest-used official name in the city's lifetime. |
| BEG_YEAR | The year in which the lifetime begins. It means that the city began to appear in this year. Its minimum value is 1368 (the year that the Ming Dynasty established), and the maximum is 1911 (the year when the Qing Dynasty ended). |
| END_YEAR | The year in which the lifetime ends. It means that the city's status changed during this year (expanding, reducing, changing the shape of the plan, or disappearing). The age range is also from 1368 to 1911. |
| TYPE | The city's administrative level in the year of the "BEG_YEAR". |
| RELIABILIT | Reliability rating of the data. |
| REFERENCES | References on which the data was mainly based. For the meaning of abbreviations, see Appendix C. |
| AREA_sq_km | Area within the city wall (unit: square kilometer). |

**Appendix B: Data records of CUED**

The China Urban Extent Dataset (CUED) in 15th-19th centuries we provide six shapefile files (referring to files with .cpg, .shp, .dbf, .shx, .xml, .sbn, .sbx and .prj extensions). It includes six

representative years (1400, 1537, 1648, 1708, 1787 and 1866). The data records of CUED in six
representative years are the same. They include the following attributes:

| FID | The (unique) identifier for each object (integer). |
|---|---|
| REP_YEAR | The representative years (i.e., 1400, 1537, 1648, 1708, 1787, and 1866). |
| NAME | City's name in the representative years. |
| TYPE | City's administrative level in the representative years. |
| RELIABLIT | Reliability rating of the data. |
| REFERENCES | References on which the data was mainly based. For the meaning of abbreviations, see Appendix C. |
| AREA_sq_km | Area of the city (unit: square kilometer). |

**Appendix C: Abbreviations**

| ACM | Guo, H., and Jin, R.: General history of administrative regions in China (the volume of Ming Dynasty), Fudan University Press, Shanghai, 2007. |
|---|---|
| ACQ | Fu, L., Lin, J., Ren, Y., and Wang, W.: General history of administrative regions in China (the volume of Qing Dynasty), Fudan University Press, Shanghai, 2013. |
| BIAM | Cheng, Y.: City wall data compilation of Book Integration of Ancient and Modern Times, China Social Sciences Press, Beijing, 2016. |
| CTW | Zhang, Z.: Ancient cities in Taiwan, Joint Publishing, Beijing, 2009. |
| LC | Cheng, Y.: City wall data compilation of local Chronicles, China Social Sciences Press, Beijing, 2016. |
| URQ | Cheng, Y.: City wall data compilation of Unified Records of the Qing Dynasty, China Social Sciences Press, Beijing, 2016. |

**Author contributions.** JX, XQ and CY originated, conceived and designed the work. CY collated
and studied the historical literatures. XQ, JX, YX and ZY developed and analyzed the dataset. All
authors contributed to the writing of the manuscript.

**Competing interests.** The authors declare that they have no conflict of interest.

**Acknowledgements.** We would like to thank Lijun Qin, Rui Sun, Shuai Cao, Yuchao Jiang,
Xiaolin Zhang of Nanjing University, Zihao Xu of Yunnan University and Xinghua Chen of
Nanjing Agricultural University for their work of the dataset.

**Financial support.** This research has been supported by the National Natural Science Foundation
of China (No.41671082).

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

**Figures and figures legends**

**Table 1.** Provincial distribution of urban extents in 15$^{th}$-19$^{th}$ centuries.

| Province | Urban extent area (km$^2$) | | | | | |
|---|---|---|---|---|---|---|
| | 1400 | 1537 | 1648 | 1708 | 1787 | 1866 |
| Anhui | 52.68 | 53.54 | 53.64 | 53.39 | 53.19 | 54.55 |
| Fujian | 40.33 | 42.04 | 43.77 | 37.88 | 38.55 | 38.71 |
| Gansu & Ningxia | 32.76 | 49.71 | 52.29 | 51.64 | 53.47 | 53.41 |
| Guangdong & Hainan | 40.26 | 44.92 | 51.32 | 49.47 | 44.05 | 44.30 |
| Guangxi | 22.34 | 23.95 | 25.46 | 24.83 | 26.24 | 26.24 |
| Guizhou | 13.08 | 14.72 | 18.34 | 15.89 | 18.18 | 18.00 |
| Hebei, Beijing & Tianjin | 168.88 | 154.87 | 182.13 | 175.69 | 180.04 | 201.36 |
| Heilongjiang | 0 | 0 | 0.29 | 5.81 | 17.53 | 18.30 |
| Henan | 102.62 | 112.01 | 113.74 | 111.26 | 112.58 | 114.32 |
| Hubei | 41.05 | 41.80 | 42.28 | 42.10 | 42.73 | 42.73 |
| Hunan | 26.85 | 26.27 | 27.70 | 26.59 | 27.26 | 27.77 |
| Inner Mongolia | 28.59 | 3.16 | 2.90 | 0.79 | 10.60 | 10.60 |
| Jiangsu & Shanghai | 122.06 | 120.26 | 127.08 | 126.27 | 127.39 | 124.55 |
| Jiangxi | 44.74 | 45.38 | 46.97 | 46.68 | 47.08 | 47.08 |
| Jilin | 0 | 0.18 | 0.18 | 4.22 | 4.68 | 5.51 |
| Liaoning | 21.34 | 26.02 | 37.73 | 37.71 | 38.93 | 39.69 |
| Qinghai | 2.23 | 2.21 | 2.66 | 2.66 | 3.03 | 3.28 |
| Shaanxi | 47.82 | 51.63 | 58.74 | 57.96 | 60.04 | 63.80 |
| Shandong | 87.22 | 92.51 | 94.80 | 93.38 | 90.56 | 104.98 |
| Shanxi | 79.68 | 91.50 | 98.37 | 97.65 | 94.13 | 93.65 |
| Sichuan & Chongqing | 55.24 | 58.71 | 59.59 | 55.30 | 58.91 | 59.72 |
| Taiwan | 0 | 0 | 0 | 3.31 | 4.03 | 4.64 |
| Xinjiang | 0.33 | 0.15 | 0.15 | 0.15 | 20.79 | 20.96 |
| Yunnan | 29.28 | 32.50 | 35.05 | 31.54 | 35.10 | 35.21 |
| Zhejiang | 82.62 | 87.44 | 87.92 | 73.91 | 74.18 | 74.41 |

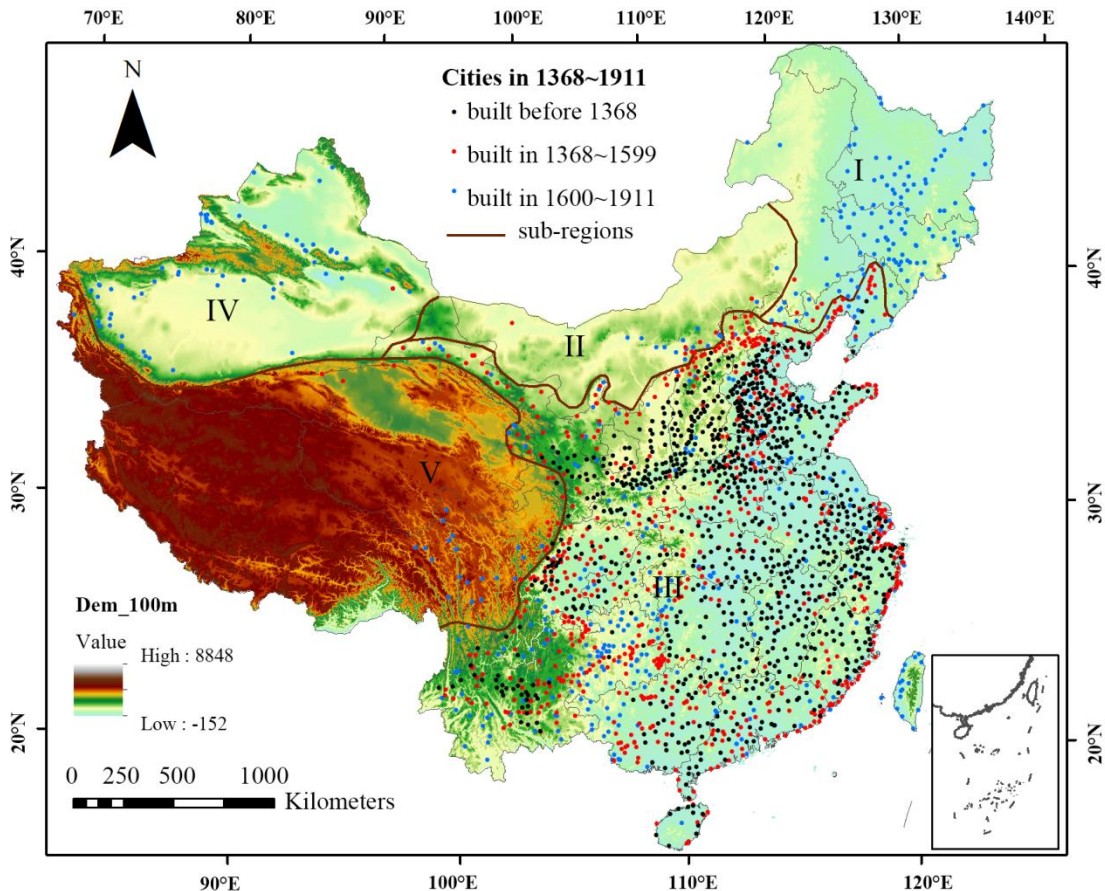

**Figure 1. Cities in the Ming and Qing Dynasties (1368-1911). The study area is divided into five natural sub-regions: Region I, Northeast China; Region II, Inner Mongolia; Region III, traditional agricultural area; Region IV, Xinjiang; Region V, Qinghai-Tibet Plateau.**

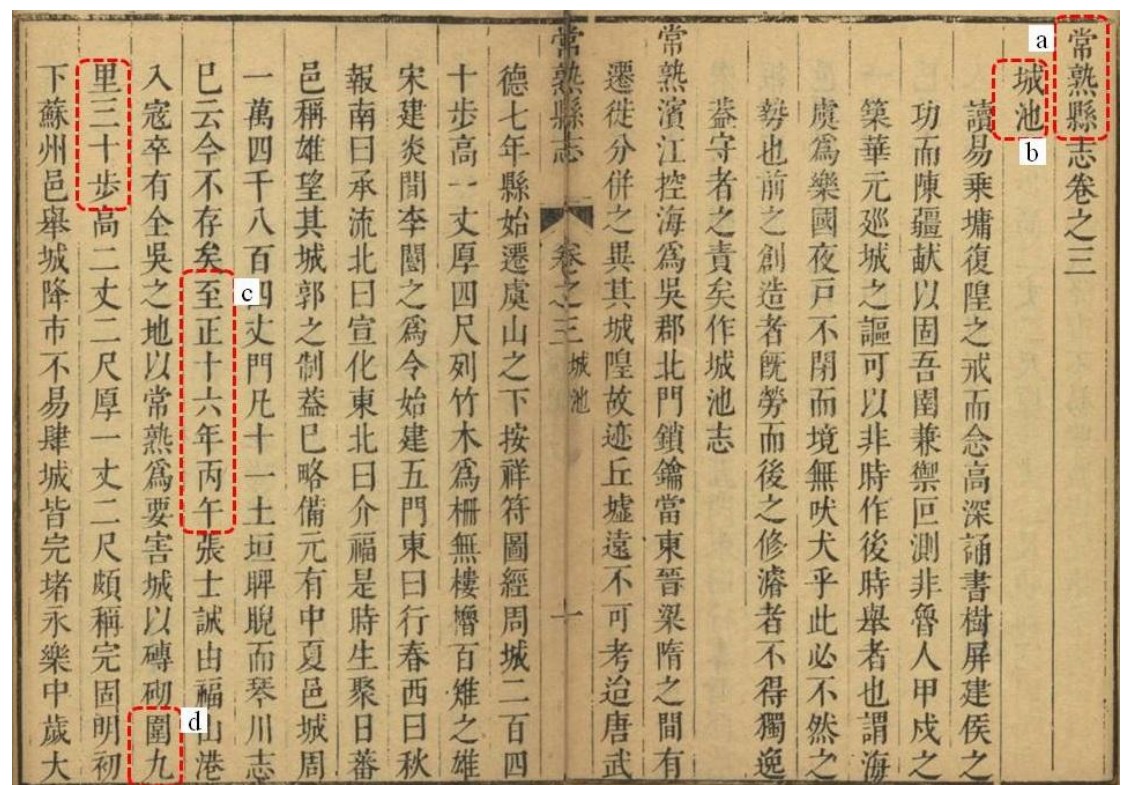

**Figure 2. The image of the record of the city wall in a *Local Chronicles* of the 17th century (*Kang-Xi Changshu county's Chronicle*). (a) City's name: Changshu (Jiangsu Province). (b) Chapter name: city wall and moat. (c) Year of the city wall built: the 16th year of *Zhizheng* in the Yuan Dynasty (1356 AD). (d) The perimeter of the wall: around 4.6 kilometers (actual about 5.44 kilometers).**

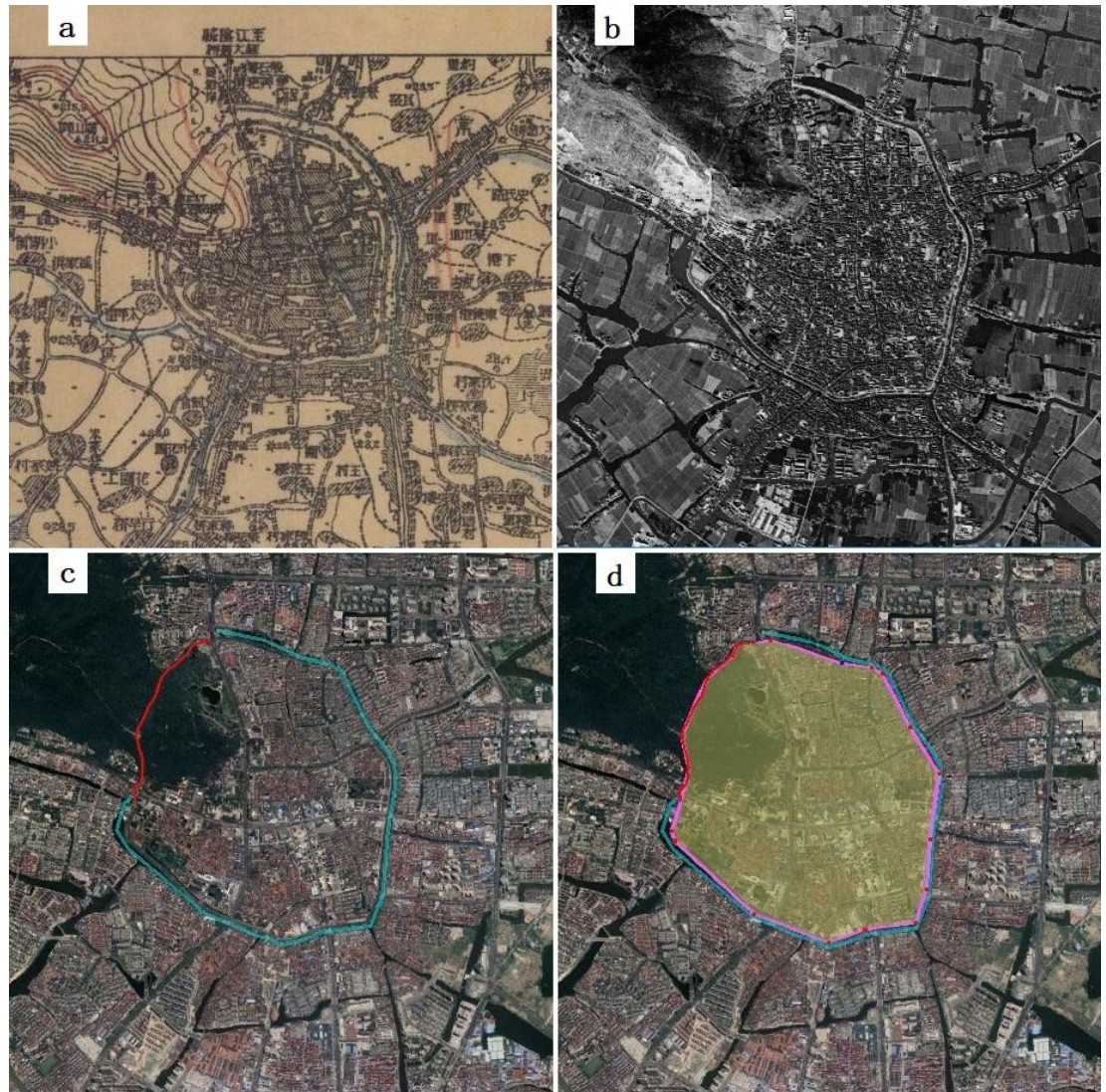

**Figure 3. Maps and remote sensing images that show the city wall and associated relics of Changshu, Jiangsu Province. (a) The 1:50,000 military topographic maps made in 1928. The jagged line on the map represents the city wall and the double line represents the river. (b) The 1970s CORONA photographs form USGS. (c) The remaining city walls (tagged as red line) and moats (tagged as blue line) are still clearly visible. The map is based on © Google Earth image, 2018. (d) According to the remains of the city walls and the moat, the scope of the city wall is drawn (yellow area). The map is based on © Google Earth image, 2018.**

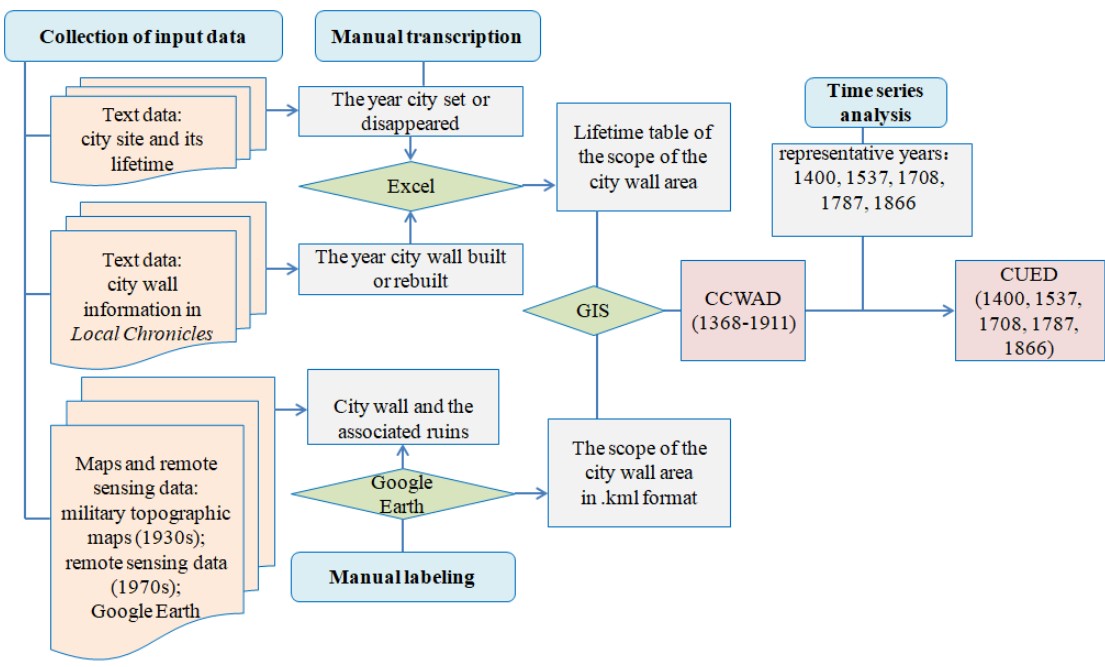

**Figure 4. A flowchart of the methodology used to generate the China City Wall Areas**
**Dataset (CCWAD) and China Urban Extent Dataset (CUED) in 15th -19th centuries in late**
**imperial China**

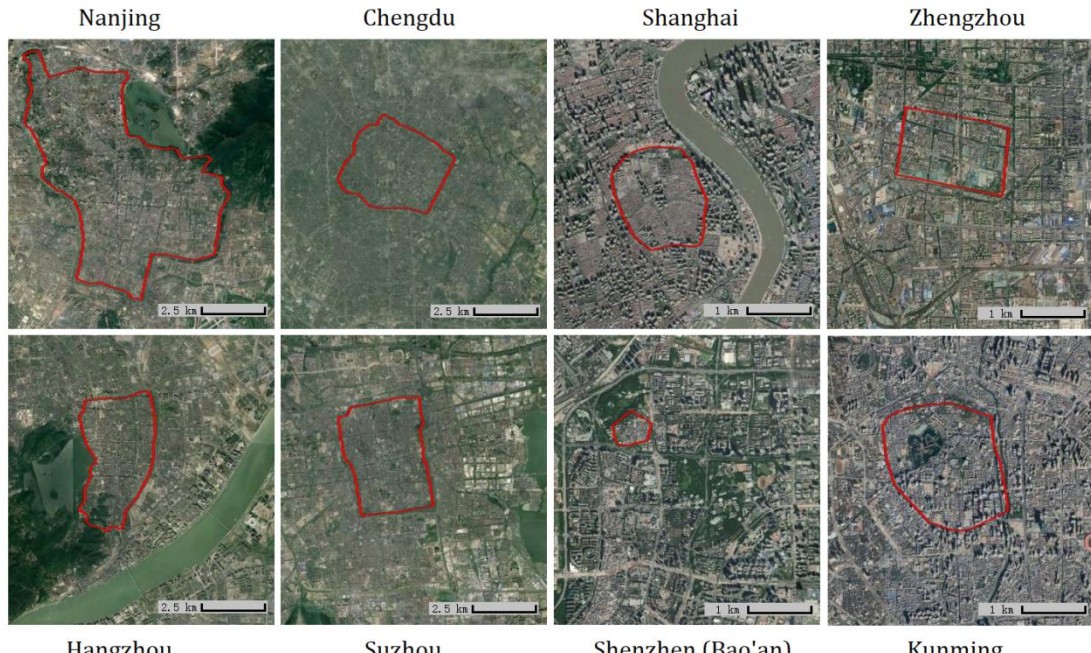

**Figure 5. Several scope of city walls of Chinese cities from 1368 to 1911. The red aerials are**
**from the China City Wall Areas Dataset (CCWAD) which illustrate the location of city walls.**
**These maps are based on © Google Earth image, 2020.**

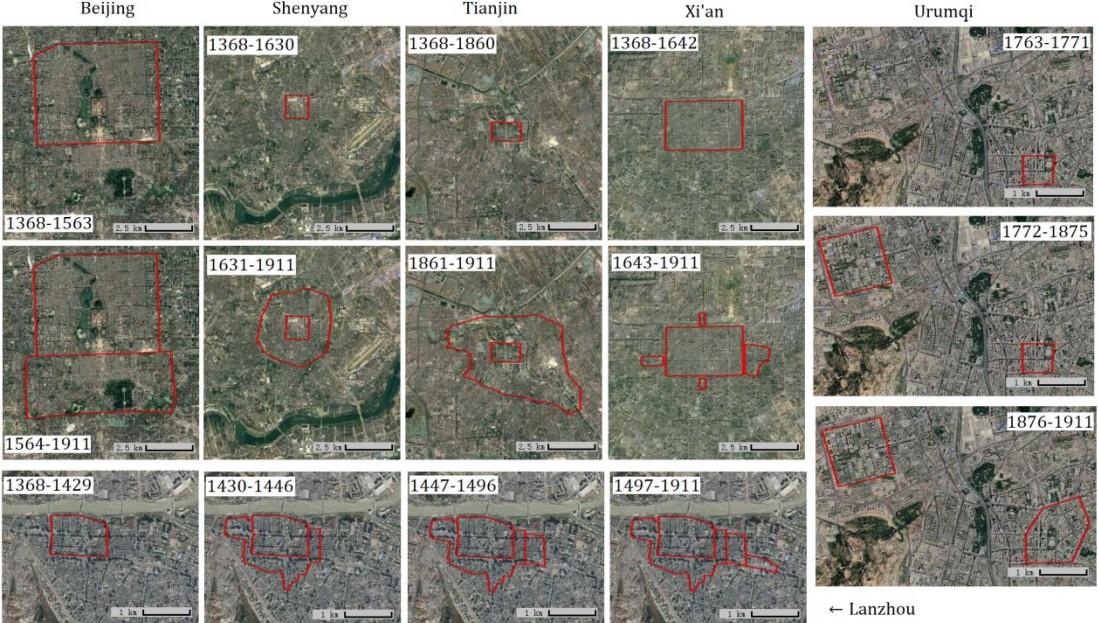

**Figure 6. Several scope of city walls of Chinese cities from 1368 to 1911. The red aerials are from the China City Wall Areas Dataset (CCWAD) which illustrate the location of city walls. These maps are based on © Google Earth image, 2020.**

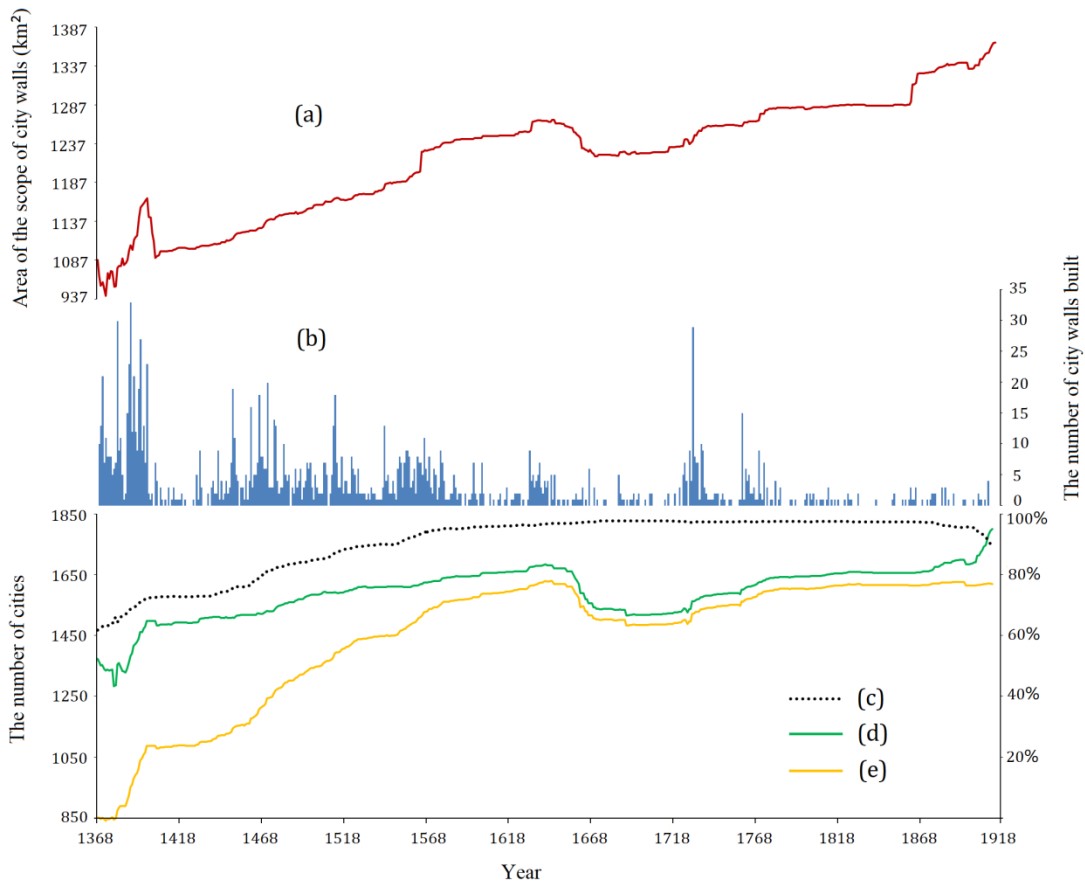

**Figure 7. Time series of cities and city walls in the Ming and Qing Dynasties (1368-1911). (a) The time series of the area of the scope of city walls. (b) The number of city walls built. (c)**

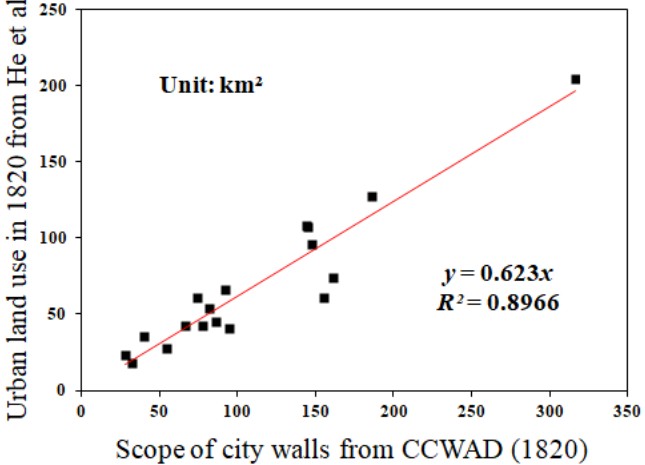

Walled cities' percentage of the total number of cities. (d) The total number of cities. (e) The

Figure 8. Comparison of the area of urban land use in 1820 (ULUD) and the area of the

scope of city walls in 1820 from CCWAD.

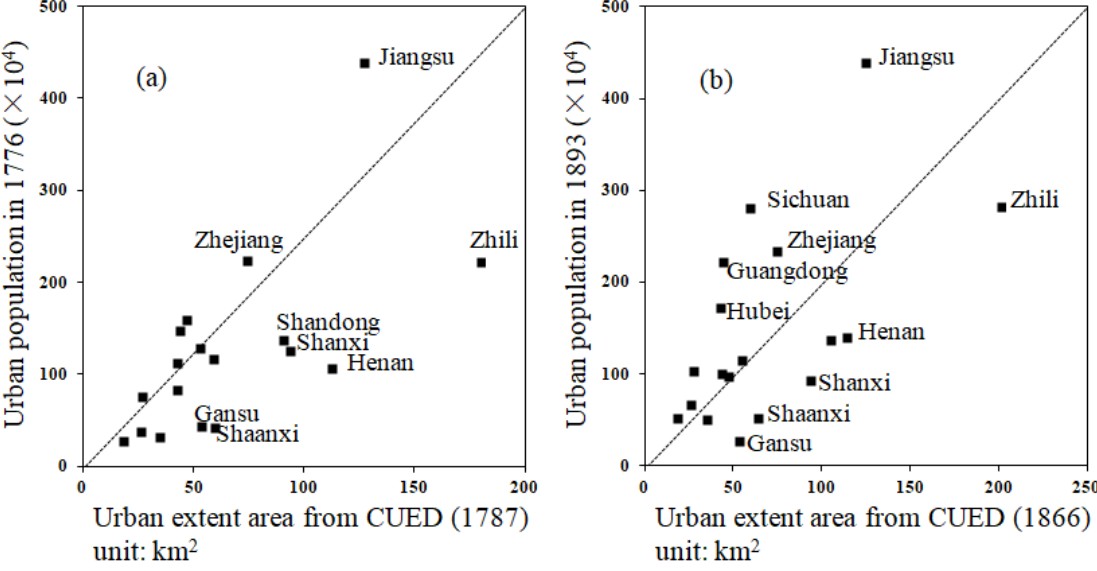

Figure 9. Comparison of the urban population in 1776 & 1893 (UPD) and the urban area in

1787 & 1866 from CUED.

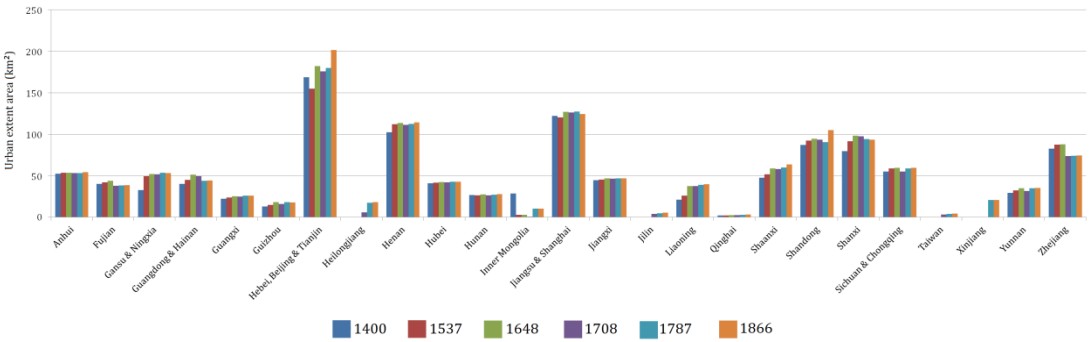

**Figure 10. Provincial distribution of urban extents in 1400, 1537, 1648, 1708, 1787 and 1866.**