# Peer review of "The dataset of walled cities and urban extent in late imperial China in 15th-19th centuries"

_Earth System Science Data, 2021_

## Author Response (AR1)

**Referee 1**

The Authors present in the data descriptor, the dataset on the urban extent in late imperial China. Although creating such datasets was definitely a time-consuming process for such a long period and large spatial extent, I have some comments, doubts, and suggestions that could help in future usage of the database, by other scholars. Please find them below.

**Response:** We thank the reviewer for valuing our work.

**Comment 1:**

The Authors write that 'most cities in late imperial China built city walls, and these walls were usually built around the urban built-up area (Yannis et al., 2017).' The paper cited here, by Yannis et al., 2017, goes much further starting with (maybe too strong – I think you can clarify that, as the specialists) quote of Osvald Sirén (1924), that 'There is no real city in Northern China without a surrounding wall, a condition which, indeed, is expressed by the fact that the Chinese use the same word ch'eng for a city and a city-wall: for there is no such thing as a city without a wall. It is just as inconceivable as a house without a roof.'

I think better clarification and support of the thesis on city walls' importance in China is needed in your paper. You state that city walls in China are the only reliable data on the urbanization processes (e.g. due to lack of other data, as stated in the paper). But it needs to be clarified because the role of city walls in China differed in the past when compared to the role of the city walls in other parts of the world at the time. Otherwise, one can assume that the dataset presented here has limited value for the scientific community.

**Response:** We thank the reviewer for the suggestion. Whether city walls were the reliable data on the urbanization process in China, is one of the key issues of our study. The reviewer's suggestion is very helpful because our discussion and references about this issue are incomplete. We will enrich the discussion in the revised manuscript and add more references. Here we shall try to describe some of the supplementary discussion.

We will attempt to summarize the city wall building history that existed in imperial China. The city wall is considered to be one of the basic symbols of ancient Chinese cities (Chang, 1986). But to be specific, cities in China were not always walled. During the 3rd to 10th centuries, even regional capital cities in China only built small-scale city walls called *Zi-cheng* (*Zi* means small and *Cheng* means city wall). The *Zi-cheng* was built around the government and military barracks, just like castles in medieval Europe. Residential areas, markets, schools and religious buildings were all outside the *Zi-cheng*. And smaller cities generally had no walls (Lu, 2011). From the 10th to 13th centuries, there were some large-scale city walls built around residential areas, but they were generally confined to few important cities. During the Mongolian-ruled Yuan Dynasty (13-14th centuries), many city walls were deliberately torn down. Only in the Ming and the Qing Dynasties (14-19th centuries), cities generally built large-scale walls to protect governments, temples, granaries, residences, and certain natural resources against invasion, tribal uprising, and peasant rebellion. According to many previous studies (Chang, 1970; Kostof, 1992; Knapp, 2000), city walls in this period were usually slightly larger than the built-up area of the city, and as the suburban areas grew, new and larger city walls were often built. Thus, the city wall in the Ming and Qing periods could be regarded as the urban fixation line, which reflected the extent of the

city. On the other hand, the Ming period and the first century of the Qing witnessed the extensive construction of city walls. As we state in lines 260-261 of the preprint, 80% of cities had walls in the 15[th] century, and in the 16[th] century, 95% of cities had city walls. Based on the brief review of the history of Chinese city walls, we believe that during the Ming and Qing Dynasties, the city wall could be regarded as a reliable data for the process of urbanization.

References:

Chang, K.: The archaeology of ancient China (4[th] edition, revised and enlarged), Yale University Press, New Haven and London, 1986.

Chang, S.: Some observations on the morphology of Chinese walled cities, Ann. Assoc. Am. Geogr., 60: 63-91, 1970.

Knapp, R.: China's walled cities, Oxford University Press, Oxford, 2000.

Kostof, S.: The city assembled: the elements of urban from through history, Thames & Hudson Ltd, London, 1992.

Lu, X.: Inside and outside the city wall: urban form and spatial structure in ancient Han River basin, Zhonghua Press, Beijing, 2011. (in Chinese).

**Changes in the revised manuscript:** Please see lines 77-98 of the revised manuscript.

**Comment 2:**

The Authors cite the works by Skinner, but do not refer broadly to the fact that Skinner produced his own database of more than 150 attributes for all cities including information on the city walls. Although in Chinese scientific literature there has been a discussion with Skinner approach (Cao 2001), I think it would be good to discuss it briefly here and refer to the main differences between the datasets, so that it is clear for the readers what is new and why it was changed, regarding previous works.

**Response:** We thank the reviewer for the suggestion. We will enrich the discussion of this subject in the revised manuscript. William Skinner used population as the key indicator to measure the urbanization of China in the 19[th] century (Skinner, 1977, page 220~221). However, since China did not have reliable urban population data until 1953, Skinner had to work backward in time, extrapolating better, more recent data to somewhat earlier dates, and building up a consistent time series culminating with the fairly hard data for 1953. He selected 1893 as the representative year, and created a comprehensive file of over 2,500 data cards designed to cover every city and town. Based on this database of more than 150 attributes (mainly including administrative level, circumference of city wall, postal status, population estimates, trade statistics and steamship or rail traffic), cities were classified. Then, he defined the urban population class intervals that the upper boundary of each class was twice the lower boundary, the following series was used: 1,000, 2,000, 4,000, 8,000, 16,000, 32,000, and so on. And finally, Skinner estimated the urbanization process of China in the 19[th] century. In summary, we think it is acceptable to use data of the 1950s to study the urbanization in the 19[th] century; but for longer-term research, the credibility and operability of this approach will be greatly reduced.

**Changes in the revised manuscript:** Please see lines 52-65 of the revised manuscript.

**Comment 3:**

Please clarify if the city walls extent was understood as administrative boundaries or where there some exceptions?

**Response:** Thank you for your comments. In most cases, the city walls had no administrative meaning, but were only functional buildings. Sharp division into distinct urban and rural civilizations seems to have vanished by the beginning of the imperial era in China (Chu, 1962; Timothy, 1985). And the administrative agency within the city wall not only managed the city, but also the countryside. Although urban administrative agencies appeared in some large capitals in 11-13[th] centuries, these agencies ceased to exist after the 14[th] century (Han et al., 2007). This may be very different from the Western European cities in the same period.

References:

Chu, T.: Local government in China under the Ch'ing, Harvard University Press, Cambridge, MA, 1962.

Han, G., Lin, Y., and Wang, C.: Cities as administrative units and the development of urban systems in the Song, Liao, Jin and Yuan Dynasties, History Research, 4: 42-62+190, 2007. (in Chinese).

Timothy, B.: The spatial structure of Ming Local administration, Late Imperial China, 6 (1): 1-55, 1985.

**Comment 4:**

In the title you have the 15th-19th centuries, although many cities from the 14th century can be also found in the database. Could you clarify that?

**Response:** Thank you for your comments. The title refers to the China Urban Extent Dataset (CUED), which contains data from only six representative years (i.e., 1400, 1537, 1648, 1708, 1878, and 1866), so it is the 15[th] -19[th] centuries. The 14[th] century cities you found are in the China City Wall Areas Dataset (CCWAD). Since CCWAD includes cities from the Ming and Qing periods, and the Ming Dynasty was founded in 1368, there are some cities from the 14[th] century in the CCWAD.

**Comment 5:**

Paragraph 6 on the data accuracy is very general and mostly based on the 'accuracy ranking', but in my opinion, it is not fully clear how did you rank the accuracy in detail? Much more information (incl. examples) is needed here since this is now the only uncertainty attribute in your dataset. Please explain the ranking rules, so that the readers have no doubts about how it was exactly done.

**Response:** We thank the reviewer for the suggestion. As you pointed out, the 'accuracy ranking' is an uncertainty attribute in our dataset. It is based on the reliability of the restored results. The accuracy ranking consists of three accuracy levels, A, B, and C, and two special case marks, D and BW. It's mainly depends on the richness of the city's historical documents and the integrity of the ground remains. But the accuracy levels are basically subjective decisions of the authors. Thus, in fact, the accuracy ranking A indicates that the authors are quite certain about the restored result, the B indicates that part of the restoration is speculative, and the C means that the restoration is based entirely on supposition. In addition, the D indicates that the city has never been walled, so its urban extent is entirely speculative. The BW expresses the speculation on the urban extent before the city built its original city wall. The hypothetical results of C, D and BW were based on the city's limited historical documents and physical remains, its administrative level as well as the size of the nearby cities. Accuracy rankings for all data were determined after discussion by all

authors.

In summary, the accuracy ranking A and B are more credible, accounting for 90% of the data of CUED, and 69% of CCWAD. The C and D together account for 5% of CUED and 17% of CCWAD. Limited by objective conditions, the extent of some cities may be difficult to restore, but it may not be appropriate to exclude these cities directly. Therefore, we developed the accuracy rankings so that users with different needs can decide how to use these speculative data. We will improve the discussion in Section 6 of the revised manuscript.

**Comment 6:**

Also, there is not much on the geometric accuracy of your data (I assume this is because it was done on recent detailed remote sensing data), but there might happened that some cities from the database were completely destroyed and are not detectable on the current remote sensing data – how was the extent of the walls assessed then? Are those cities present in the database or not?

**Response:** Thank you for your comments. Yes, these cities present in our database. The restoration results of these cities were entirely based on speculation, so we marked them with 'C' in the 'accuracy ranking' attribute. There are 269 of these data in CUED, accounting for 3%; and 140 in CCWAD, accounting for 4%.

**Comment 7:**

The datasets' attributes are explained in the Data Records files, but there are discrepancies between the attributes presented there and in the shapefiles. E.g. in CEUD dataset, we can find the attribute 'TYPE' which is not explained in the Data Records file. Similarly, in CCWAD attributes we can find 'area', but Data Records do not clarify the unit it is presented in. Attribute 'References' should be fully explained – e.g. in the form of the full list of options which can be found there. Preferably the list of attributes should be explained both in the Data Descriptor (e.g. in the Data availability section or in the form of a respective table) and in the Data repository.

**Response:** Thank you very much for pointing out this. These are mistakes in our work. We will correct these problems in the revised dataset (see https://doi.org/10.6084/m9.figshare.14112968.v2) and add supplementary discussion in the revised manuscript. The attribute texts were simplified when converting from excel to shapefiles. So we will develop an abbreviations table of the attribute 'References' and add it as Appendix C in the revised manuscript.

**Changes in the revised manuscript:** Please see Section 8 and Appendix A, B, and C of the revised manuscript.

**Comment 8:**

It is not clear why in some cases there are differences among the databases and cities covered. For instance, in the CCWAD there is a city of Yijinai which is attributed by the end year 1372, while in CUED it is covered only for the representative year 1400, but as Weiyuan. None of the later periods cover it. Is it because it changed the name, the extentt, and did not survive till the next, 1537 (or any later) representative years?

**Response:** Thank you for your comments. Yes, it is. According to Guo and Jin (2007), the city Yijinai was built before the Ming period. The Ming army occupied the city in 1370 and abolished the administrative agencies of the city in 1372. Then, around 1384, a garrison was set up at this site, and its name was changed to Weiyuan. And around 1403, the garrison was abolished, and the

city was abandoned too. So in CUED, the city present in 1400, but not in 1537 or any later.

References:

Guo, H., Jin, R.: General history of administrative regions in China (the volume of Ming Dynasty), Fudan University Press, Shanghai, page 87, 393~399, and 410~411, 2007. (in Chinese)

**Comment 9:**

The above-mentioned example shows an important issue of the database – in the Data Records CCWAD file you write: 'Due to the ancient Chinese cities often have several names at the same time and they are always change, here we provide the most common name for them.' -it is not clear what 'most common' mean. I would recommend reconsidering that issue – would it be possible to either present all the names as the attributes (I suppose you have it for representative years) or use e.g. clear rule like, the last name, the name which was used for the longest period. You can also explain better 'most common', as now it is not fully clear.

**Response:** Thank you very much for pointing out this. The names of ancient Chinese cities changed frequently, and cities often had several names at the same time. However, city name is not the focus of this study, because the China Historical Geographic System (CHGIS) developed by Harvard University and Fudan University has provided an excellent database including historical city name. So we simplified this issue when developing the CCWAD dataset. However, as you pointed out, 'most common' is really not an appropriate naming rule. So we decided to change the naming rule to the longest-used official name of the city in its 'lifetime' (from its 'begin year' to the 'end year'). We will revise the city names in the updated CCWAD dataset according to the new naming rule (see https://doi.org/10.6084/m9.figshare.14112968.v2).

**Changes in the revised manuscript:** Please see the 'Data Records' of the revised dataset and Appendix A of the revised manuscript.

**Comment 10:**

The Authors write that: 'In addition, we also need some remote sensing images for auxiliary work. The 1970s China remote sensing image from the U.S. Geological Survey (USGS) website (https://earthexplorer.usgs.gov/) was the most important'. It is not explained in the text or in Figure 3b (or l. 200), which sensors exactly were used here? We assume it was not Landsat, so please state clearly which images were used and add respective details. This is important because actually, remote sensing data are the main source of the extent of the walls in your work.

**Response:** Thank you very much for pointing out this. We used CORONA photographs here. CORONA is the satellite deployed by the United States in 1958, and it takes remote sensing images covering the world from 1960 to 1972. Now the CORONA photographs have been decrypted and can be downloaded from the USGS website (https://earthexplorer.usgs.gov/). We will clarify these in the revised manuscript.

**Changes in the revised manuscript:** Please see lines 168-174, 228-230, and 507 of the revised manuscript.

**Comment 11:**

Is there a possibility to verify somehow, e.g. for selected cities, if your extent is assessed correctly, e.g. by comparing the area to the reliable statistical records from the respective period?

**Response:** We thank the reviewer for the suggestion. As you pointed out, our dataset lacks

comprehensive validation. We think it is because this part of the research is inductive and based on experience. All the reliable urban records (such as historical literatures, old maps and remote sensing image) that we can find are collected and used in the restoration work based on the historical urban morphology theory. And the restored results are derived from the summary of various records, rather than from hypotheses or prior models. So it may be difficult to find a suitable validation method. In order to remedy the lack of verification of the dataset, we designed the 'accuracy rankings' so that users with different needs can decide how to use these data. Please refer to Section 6 of the revised manuscript.

**Comment 12:**

40-42 – how this work can help in Chinese sustainable urbanisation currently? It is somehow unclear (similarly the reference to sustainable development goals in the Abstract).

**Response:** We thank the reviewer for the suggestion. We mainly consider that the cores of current Chinese cities are mostly developed on the basis of ancient times. These areas are generally located in the center of cities, and most of them are still densely populated and flourishing businesses. And the old cities concentrate a large number of cultural heritages and landscape. So the preservation and renewal of old cities is a hot issue in urban development in China. Our work provides basic dataset on the evolution of the old cities during the Ming and Qing dynasties, and may be helpful to researchers who are interested in the sustainable development of cities. Of course, that's not the subject of our study, and we don't mind deleting these unimportant expressions.

**Changes in the revised manuscript:** Please see the Abstract and line 40 of the revised manuscript.

**Comment 13:**

125-126 – the same title appears twice – was it a different publication or edition? Please clarify.

**Response:** Thank you very much for pointing out this. It is a clerical error. It should be 'the *Book Integration of Ancient and Modern Times* and *Unified Records of the Qing Dynasty*'. And lines 118-120 should be 'the *Book Integration of Ancient and Modern Times* (edited in 1701-1728), *Unified Records of the Qing Dynasty* (edited in 1842)'. We will correct them in the revised manuscript.

**Changes in the revised manuscript:** Please see lines 146-147 and 152-153 of the revised manuscript.

**Comment 14:**

There are some minor English spelling and grammar issues (e.g. line 100 – 'lat' instead of 'late') and repetitions (line 110-114 – word 'region' is repeated 5 times).

**Response:** Thank you very much for pointing out this. We will correct them in the revised manuscript.

**Changes in the revised manuscript:** Please see lines 128 and 139-142 of the revised manuscript.

**Comment 15:**

Style should also be corrected – e.g. lines 240-241, l. 251 – 'It should to analysis the time series…', l. 339-340.

**Response:** Thank you very much for pointing out this. We will correct them in the revised manuscript.

**Changes in the revised manuscript:** Please see lines 267-268, 287-288, and 387-388 of the revised manuscript.

**Comment 16:**

The Authors use the names of the regions as Region I, II, etc – did you consider replacing it with proper geonames?

**Response:** We thank the reviewer for the suggestion. We consider replacing Region I to V with Northeast China, Inner Mongolia, traditional agricultural area, Xinjiang, and Qinghai-Tibet Plateau. We will label them in the revised manuscript.

**Changes in the revised manuscript:** Please see lines 493-494 of the revised manuscript.

**Comment 17:**

In the Conclusion the Authors explain a very interesting relation between city walls and war – could you elaborate it? It might be very important in terms of your dataset usage.

**Response:** We thank the reviewer for the suggestion. Like most ancient civilizations, city walls in China were primarily defensive military structures. In peacetime, the city walls were useless and often hindered the expansion of cities. During these periods, suburbs grew outside the city gates, and the walls were often neglected or even vandalized. But during the war, the walls became necessary facilities to defend the cities. At this time, if the suburbs outside the city gates had grown large, new suburban walls were built to protect them. As can be seen from Figure 7b, the peak periods of city walls construction almost without exception corresponded to all the important wars in China in the $14^{th}$ to $18^{th}$ centuries: (1) from 1368 to the early $15^{th}$ century, it corresponded to the wars that the Ming destroyed the Yuan Dynasty and civil war of the Ming Dynasty; (2) from the middle of the $15^{th}$ century to the middle of the $16^{th}$ century corresponded to the protracted wars between the Ming Dynasty and Mongols and Japanese pirates; (3) the first half of the $17^{th}$ century corresponded to the late Ming peasant uprising and the wars with the Qing Dynasty; (4) and the first half of the $18^{th}$ century corresponded to the wars in Xinjiang. It can be seen that the construction of city wall was closely related to the stimulation of war. The development of cities generally required peaceful social environment, but city walls were often built during wartime. In this sense, the city wall can be seen as the sign and confirmation of the urban development before wars. We will add relevant discussion in the Conclusion.

**Changes in the revised manuscript:** Please see lines 420-432 of the revised manuscript.

**Comment 18:**

In point 3 (l. 379) of the Conclusions the Authors mention the cities without the walls – could you explain, how common the phenomena was? When and why it happened?

**Response:** We thank the reviewer for the suggestion. We use accuracy ranking D to represent the cities without walls in CUED and CCWAD. In CCWAD, there have 436 such kind of cities, accounting for 13%. In CUED, there are 83 such cities in the representative year 1400, 48 in the year 1537, 43 in the year 1648, 31 in the year 1708, 37 in the year 1787, and 42 in the year 1866; and the proportions are between 2% and 5%. It can be seen that there were not many cities without the walls. Cities without the walls could be roughly divided into two categories. One was the less

important cities located in the inland areas with better security. The other was the cities established at the end of the 19$^{th}$ century. At that time, with the advancement of weapons, the defensive significance of the city wall was greatly reduced. We will add relevant discussions in the Conclusion.

**Changes in the revised manuscript:** Please see lines 435-443 of the revised manuscript.

**Referee 2**

Records of urban extent are useful for not only investigating the urban development, especially for China, which has a long history of urban activate as well as among one of the fastest urban development hotspots worldwide. Many urban extent datasets have been produced in the past decades, benefited by the unprecedented capability of both Earth observation and machine learning; however, these datasets mainly focus on delineating the urban-related land covers, e.g., impervious surface, for the past 4 decades or so, due to the relatively short history of Earth observation. Extending the records deeper into the history of urban extent would provide a more complete picture of the urban development. Thus, I believe that datasets with an older/longer history of urban development would provide valuable information for complementing the urban records provided by the other urban land cover dataset. However, I think there are still flaws in both the manuscript and the presented dataset. These need to be addressed before the manuscript to be published.

**Response:** We thank the reviewer for valuing our work.

**Comment 1:**

The authors did not provide a clear definition of "urban" in the manuscript, which is commonly considered as "a primary nexus of human and environmental system interactions". The authors claimed to provide an urban extent dataset, but what it actually providing is the extent of the city wall. The definition gaps between the city wall and urban extent are considerable. Although the authors explain the relationship between these two, the connection is still too weak, and further clarifications would be needed. Can a city wall equal or able to represent the actual urban extent? The construction of the city wall would have taken decades to complete, and the extent of the wall could considerably lag behind the changes of urban extent. I can agree that the main urban activities were inside the city wall, but I don't believe it's completely within it, especially during peaceful and economically prosperous times. Both Ming and Qing dynasties had large territories and enjoyed long peaceful and prosperous periods, particularly for areas away from the border. Representing the urban extent with only the extent of city wall would significantly underestimate the actual urban extents for many cities in the two dynasties. The authors argued in the manuscript that the city wall has been used to study the extent of cities, and I think it could be reliable for some individual cities, but I could not agree that the city wall can be used as the indicator for all cities.

**Response:** We thank the reviewer for the suggestion. And we think your opinions are very crucial. How can the scope of a city wall represent the actual urban extent, is one of the key issues of this research. Our discussions on this issue are clearly inadequate, and we will make further clarifications in the revised manuscript.

We believe that although the construction of the city wall took a long time, it was a functional building with high cost. And it would be built only when it was of vital importance to military and economic defense. Therefore, the scope of the city wall must be adapted to the physical boundaries of the urban built-up area at that time. But on the other hand, as you pointed out, the urban extent would not remain unchanged forever, it would change accordingly with the increase of decrease of urban residents. For example, the overflowing population would build contiguous settlements outside the city wall, especially during periods of peaceful and prosperous periods.

And during these periods, the scope of city wall could not be consistent with the actual urban land use. After comprehensive consideration, we think that the city wall could be regarded as the urban boundary at least during the period when the city wall exerts its functional role; and the closer the time to the construction of the city wall, the more consistent the scope of city wall and the urban extent. Therefore, as long as the appropriate periods were selected, the scope of city walls in these periods could be very approximately regarded as the urban extent. For discussion about these issues, please see lines 77-98 and 267-288 of the revised manuscript.

Therefore, we provide two datasets, namely the China City Wall Areas Dataset (CCWAD) and the China Urban Extent Dataset (CUED). Firstly, we produced the CCWAD, which reflected the scope of the city walls in China during the Ming and Qing Dynasties. After that, we studied the history of the city wall construction in the Ming and Qing Dynasties, and selected six representative years (i.e., 1400, 1537, 1648, 1708, 1787, and 1866) from the peak period of city wall construction to generate the CUED, which reflected the urban extent in the six representative years. For the process, methods and principles of the above works, please refer to lines 289-311 of the revised manuscript.

**Comment 2:**

The cities wall were mainly built for large cities. Could it lack representation for smaller cities or towns?

**Response:** We thank the reviewer for the suggestion. According to this study, cities in the Ming and Qing periods generally built city walls (see lines 433-443 of the revised manuscript). Our dataset contains 2,376 cities, most of which are smaller cities such as county seats. These cities, as we described in lines 294-299 of the revised manuscript, 80% of them walled in the $15^{th}$ century, and 95% of them walled in the $16^{th}$ century. So we think that in this study, the representation of smaller cities should be sufficient.

**Comment 3:**

The dataset has no validation. I understand that validation for such a historical record is difficult, but I am afraid that it cannot be accepted without comprehensive validation. Nevertheless, I think there could be still ways to assess its quality, for example, evaluating the changes of the extents for particular regions with reliable urban records?

**Response:** We thank the reviewer for the suggestion. As you pointed out, we did not find a comprehensive validation method. All the reliable urban records (such as historical literatures, old maps and remote sensing image) that we can find are collected and used in the restoration work based on the historical urban morphology theory. This part of the research is inductive and based on experience. And its results are derived from the summary of various records, rather than from hypotheses or prior models. So it may be difficult to find the suitable validation method. In order to remedy the lack of quality verification of the dataset, we designed the 'accuracy rankings' so that users with different needs can decide how to use these data. Please refer to Section 6 of the revised manuscript.

**Comment 4:**

Line 241-248, the authors explained the challenges of using the city wall for representing the urban extent. However, the manuscript did not carry out a convincing solution to address those.

**Response:** We thank the reviewer for the suggestion. Our discussions on this issue are clearly inadequate, and we will make further clarifications in the revised manuscript. We think that if the appropriate periods were selected, the scope of city walls in these periods could be regarded as the urban extent. This is why we provide two datasets, namely the CCWAD and the CUED. And the CUED only reflects the urban extent in the six representative years (i.e., 1400, 1537, 1648, 1708, 1787, and 1866). We believe that in these six representative years, the scope of the city wall is highly consistent with the urban extent. For the process, methods and principles of these work, please refer to lines 289-311 of the revised manuscript.

**Comment 5:**

Line 38, "did not start slowly". Please remove "slowly".

**Response:** Thank you very much for pointing out this. We will correct it in the revised manuscript.

**Changes in the revised manuscript:** Please see line 38 of the revised manuscript.

**Comment 6:**

Line 150-151, why a city is considered a new city when it is chosen as an administrative center? What if the city was already there, and got chosen later to become an administrative center?

**Response:** We thank the reviewer for the comments. This is because the objects of our study only include administrative cities. Lines 122-125 of the revised manuscript defined the 'city' of this study. So 'if the city was already there, and got chosen later to become an administrative center', in this case, data before the 'city' became the administrative center were not included in our datasets. This definition of city obviously excludes some settlements that should be regarded as cities, but we haven't made enough clarification in the preprint. We will make further explanation in the revised manuscript. Thank you very much for pointing out this.

Although almost all cities in the late imperial China can be classified as administrative cities, we must point out that the following types of settlements can also be regarded as 'cities', but they are not included in our datasets. (a) In the late imperial China, the industrial and commercial settlements without administrative agencies were generally called 'markets (*shi*)' or 'towns (*zhen*)'. The size of the town was generally smaller than the lowest administrative center, the county seat. But there were also some huge towns, such as Hankou, Foshan, and Jingde, etc., whose scale exceeded the county seat and even higher-level cities. These huge towns should undoubtedly be regarded as cities, but they are not in scope of this research. (b) If a city was already there, and got chosen later to become an administrative center, in this case, data before the 'city' became the administrative center were not included in our datasets. (c) Cities outside the direct administration of the Ming and Qing empires, such as Lhasa. (d) Cities belonging to colonists, such as Macau, Hong Kong, and Qingdao, etc. The definition of 'city' or 'urban' in the late imperial China is complex and far from conclusive, but we hope that the content of our datasets to have a clear border. Therefore, in this study, we defined "city" as the settlement which the administrative center was located. And this definition is the same as the general research practice of pre-modern China.

**Changes in the revised manuscript:** Please see lines 444-460 of the revised manuscript.

**Comment 7:**

Line 166-168, The sentence does not make sense, please rephrase.

**Response:** Thank you very much for pointing out this. We will correct it in the revised manuscript.

**Changes in the revised manuscript:** Please see lines 194-196 of the revised manuscript.

**Comment 8:**

Line 226, why transform the data? Transform from what to what? Please provide more details.

**Response:** Thank you very much for pointing out this. This is a clerical error, and we will correct it in the revised manuscript.

**Changes in the revised manuscript:** Please see line 254 of the revised manuscript.

**Comment 9:**

Please clarify why the representative years were so precisely selected? How did the authors make sure the years, for example, 1648, and how all the cities had updated records for the exact year?

**Response:** We thank the reviewer for the comments. The procedure for selecting the six representative years is as follows. Firstly, we analyzed the time series of cities and city walls in the Ming and Qing Dynasties (Figure 7a-b), and divided them into six time periods (i.e. 1368-1404, 1405-1564, 1565-1662, 1663-1727, 1728-1860 and 1861-1911). Then we selected the representative year from each time period. The selection criteria are as follows. Firstly, the proportion of cities with walls in the total cities should be higher. The proportion should generally be more than 90%, except in the 14th and early 15th centuries. Secondly, after the city walls were built, the scope of the city walls generally did not change with the built-up areas over time, so the representative years should be within only one or two years after the end of a large-scale construction activities. In addition, the representative year should be selected at a moderate level of changes in the scope of the city wall within the period. Finally, the representative year should avoid major political, military events and severe natural disasters in order to reflect the general level of urban development in that period. Please see lines 289-311 of the revised manuscript.

**Comment 10:**

Line 202-203, Google Earth is a tool, not a platform.

**Response:** Thank you very much for pointing out this. We will correct it in the revised manuscript.

**Changes in the revised manuscript:** Please see line 229 of the revised manuscript.

**Comment 11:**

Line 256, what's the implication of the correlation?

**Response:** We thank the reviewer for the comment. We think that the word 'correlation' may not an accurate expression. There is only some connection between the number of wall constructions and the area of the walls scope, that is 'the periods of more constructions were often of faster area growth, and the less construction periods were always of area decline or unchanged' (lines 292-294 of the revised manuscript). We will make improvements in the revised manuscript.

**Changes in the revised manuscript:** Please see line 291 of the revised manuscript.

**Comment 12:**

Line 268, please clarify how the proportion of cities were calculated? What's the number of total cities?

**Response:** We thank the reviewer for the comment. The total number of all cities (figure 7d) and the total number of walled cities (figure 7e) were obtained through annual query in the CCWAD dataset. The walled cities' proportion of all cities (figure 7c) was obtained by dividing the number of walled cities in each year by the total number of cities in the year.

**Comment 13:**

Line 296-297, please rephrase the sentence to fix grammar errors.

**Response:** Thank you very much for pointing out this. We will correct it in the revised manuscript.

**Changes in the revised manuscript:** Please see lines 344-346 of the revised manuscript.

**Comment 14:**

Line 298, add "a" before "slow rate".

**Response:** Thank you very much for pointing out this. We will correct it in the revised manuscript.

**Changes in the revised manuscript:** Please see line 346 of the revised manuscript.

**Comment 15:**

Line 299, it would be odd to describe years using minimum and maximum.

**Response:** Thank you very much for pointing out this. We will correct it in the revised manuscript.

**Changes in the revised manuscript:** Please see lines 344-346 of the revised manuscript.

**Comment 16:**

Line 300-302, the periods were already explained in the previous sections.

**Response:** Thank you very much for pointing out this. We will remove the duplicate sentences.

**Changes in the revised manuscript:** Please see line 349 of the revised manuscript.

**Comment 17:**

Line 304 and 311, is the building of military cities actually related to urban development?

**Response:** We thank the reviewer for the comment. Yes, it is. By querying the CCWAD, we can see that the newly added cities at the end of the 14th century were mainly military cities (their type were 'Wei' and 'Suo'). These military cities were distributed along the Great Wall, the southeast coast and the southwest regions that were unstable in the early Ming Dynasty. The abandoned cities in the mid-17th century were also mainly military cities. So the building of military cities was actually related to urban development.

**Comment 18:**

Line 324-325, the regions have long development but the city walls did not expand. Does it mean the city wall lagged behind while the urban experienced development?

**Response:** We thank the reviewer for the comment. These areas (Anhui, Guangxi, Hubei, Hunan, Jiangxi, Sichuan and Chongqing) were mainly agricultural areas in the Ming and Qing period, and

were not areas with rapid industrial, commercial and urban development. So we are more inclined to think that the dataset correctly reflects the history of slow urban development in these regions during the 15th-19th centuries.

---

## Author Response (AR2)

**Report #2**

The manuscript has been greatly improved from the revision. Most of the questions have been addressed in my opinion. As stated in the previous review comments, the research could be valuable for presenting the changes of the city walls in China, and possibly helpful information for understanding the urban changes in the past centuries in China. I still have few concern regarding the current state of the manuscript.

**Response:** We thank the reviewer for valuing our work.

First, the authors claim the dataset to be about urban changes, but it is really about city walls. Although city wall could be an helpful indicator for representing the extent of cities, there are always gaps and latencies in both definitions and spatiotemporal changes between the city walls and urban extents. Most importantly, the urban dataset introduced by the manuscript is pretty much the same as the city wall dataset except presented differently. In this case, the value of the urban dataset over the city wall dataset would be very insignificant. Also, it does not make sense to have so many cities remain the exactly same urban extent over past 5 centuries. Overall, it seems misleading to claim the dataset to be an urban dataset.

**Response:** Thank you for your comments. As you pointed out, on the one hand city wall could be a helpful indicator for representing the urban extents, but on the other, there are many differences between them. We have discussed this issue in paragraphs 1-2 of section 5. Our main view is that "the city wall could be regarded as the urban boundary at least during the period when the city wall exerts its functional role; and the closer the time to the construction of the city wall, the more consistent the scope of city wall and the urban extent" (lines 280-282). Your comments remind us that this view is one-sided. This view illustrates the relationship between city wall and the urban extent of a single city on a small-scale, but on a national-scale, it is impossible for all cities in the country to build city walls at the same time. So as a national-scale dataset, the value of CUED seems to be insignificant. And the CCWAD is sufficient for the walls and urban extents of each city.

However, long-term and large-scale urban extent data are highly desirable for urban studies. Since city wall can be regarded as a helpful indicator of the extent of cities, we still hope to provide some acceptable large-scale urban extent data with long period. And the CUED is such a product. Users can certainly choose the years they need in CCWAD, such as 1400, 1500, 1600, 1700 and so on. But we try to reduce the gaps and latencies in spatiotemporal changes between the city walls and urban extents by selecting some appropriate representative years. It reduces the accuracy, but it does expand the scale. CUED attempt to find a balance between the scope of city walls, long-term and large-scale, so as to provide some acceptable and user-friendly urban extent data. And this is the meaning and value of CUED.

Thank you for your comments to make us realize that our clarification is not enough. We sincerely accept your comments that it seems misleading to claim the dataset to be an urban dataset. We have overemphasized the meaning of CUED, but in fact it is just a derivative of the city wall dataset. So it is necessary to make further improvement to the manuscript. Firstly, we consider changing the title of the manuscript and the dataset to "The dataset of walled cities and urban extent in late imperial China in 15$^{th}$-19$^{th}$ centuries". In addition, we will make a series of clarification in the abstract and main text of the revised manuscript. Please see the abstract and

lines 106-111, 145, 262-267, 284-291, 318-321, 324-325, 440-443, and 449 of the revised manuscript. It is hoped that the improved manuscript can better illustrate the significance and limitation of our dataset.

Second, accuracy is always the most important part of a data description paper. Although it is understandable that the difficulty for evaluating the accuracy of such an dataset is high, I am not sure it could be acceptable for publishing it without an accuracy evaluation. The authors did provide a ranking result and adopted it as accuracy assessment. However, it is only an internal quality flag, which can hardly be considered as an accuracy evaluation. If the authors claim the dataset described by the manuscript is about urban, then, in my opinion, it has to be properly accessed by referring to independent datasets that reflecting urban and its changes. If the authors cannot find historical urban records, other datasets, such as reliable population records available for certain regions, can also be considered for evaluating the results by examining their correlation.

**Response:** We thank the reviewer for the suggestion. We have found provincial urban land use area and urban population records in the Qing Dynasty to evaluate our results. Please see lines 322-323, 351-383 and figure 8-9 of the revised manuscript.

Third, the sub-region definitions are inconsistent between these in the main text (line 133-137) and caption of Figure 1.

**Response:** Thank you very much for pointing out this. We have corrected it and please see lines 132-140 of the revised manuscript.

Four, there are still grammar errors and wording issues. For example:

Line 25, change "earth" to "Earth".

**Response:** We have corrected it and please see line 25 of the revised manuscript.

Line 56, change He to Skinner.

**Response:** We have corrected it and please see line 56 of the revised manuscript.

Line 176, change "the amount of" to "information of".

**Response:** We have corrected it and please see line 175 of the revised manuscript.

Line 75, change "the scope city walls" to "the scope of city walls".

**Response:** We have corrected it and please see line 75 of the revised manuscript.

Line 177, "when they disappeared contributes"? Not sure why the word "contributes" is here.

**Response:** Thank you very much for pointing out this. This is a clerical error. We have corrected it and please see line 176 of the revised manuscript.

Line 195, what does "urban form space" refer to?

**Response:** Thank you very much for pointing out this. This is a clerical error. We have corrected it and please see line 194 of the revised manuscript.

Line 248-249, duplicated "the".

**Response:** Thank you very much for pointing out this. We have corrected it and please see line 247 of the revised manuscript.

Line 270, "increase of" change to "increase or".

**Response:** Thank you very much for pointing out this. Please see line 271 of the revised manuscript.

Line 349, Is it supposed to be 1368 instead of 1369?

**Response:** Thank you very much for pointing out this. Yes, it is. We have corrected it and please see line 391 of the revised manuscript.

The abbreviation CUED already has the word "dataset" in the name, it would be duplicated to mention it as "CUED dataset". For example, at line 398.

**Response:** Thank you very much for pointing out this. We have corrected it and please see line 440 of the revised manuscript.

Section 8, the authors mixed the citing of Appendix A and B.

**Response:** Thank you very much for pointing out this. We have corrected it and please see line 438 of the revised manuscript.